# Similarity-Navigated Conformal Prediction for Graph Neural Networks

**Jianqing Song[1],  Jianguo Huang[2,3],  Wenyu Jiang[2,1],  Baoming Zhang[1],**
**Shuangjie Li[1],  Chongjun Wang[1]***

[1]State Key Laboratory of Novel Software Technology, Nanjing University
[2]Department of Statistics and Data Science, Southern University of Science and Technology
[3]College of Computing and Data Science, Nanyang Technological University

## Abstract

Graph Neural Networks have achieved remarkable accuracy in semi-supervised node classification tasks. However, these results lack reliable uncertainty estimates. Conformal prediction methods provide a theoretical guarantee for node classification tasks, ensuring that the conformal prediction set contains the ground-truth label with a desired probability (e.g., 95%). In this paper, we empirically show that for each node, aggregating the non-conformity scores of nodes with the same label can improve the efficiency of conformal prediction sets while maintaining valid marginal coverage. This observation motivates us to propose a novel algorithm named *Similarity-Navigated Adaptive Prediction Sets* (SNAPS), which aggregates the non-conformity scores based on feature similarity and structural neighborhood. The key idea behind SNAPS is that nodes with high feature similarity or direct connections tend to have the same label. By incorporating adaptive similar nodes information, SNAPS can generate compact prediction sets and increase the singleton hit ratio (correct prediction sets of size one). Moreover, we theoretically provide a finite-sample coverage guarantee of SNAPS. Extensive experiments demonstrate the superiority of SNAPS, improving the efficiency of prediction sets and singleton hit ratio while maintaining valid coverage.

## 1  Introduction

Graph Neural Networks (GNNs), which process graph-structured data by the message-passing manner (Kipf and Welling, 2017; Hamilton et al., 2017; Velickovic et al., 2018; Xu et al., 2019), have achieved remarkable accuracy in various high-stakes applications, e.g., drug discovery (Li et al., 2022), fraud detection (Liu et al., 2023) and traffic forecasting (Jiang and Luo, 2022), where any erroneous prediction can be costly and dangerous (Amodei et al., 2016; Gao et al., 2019). To improve the reliability of prediction results, many methods have been investigated to quantify the model uncertainty (Gal and Ghahramani, 2016; Guo et al., 2017; Kendall and Gal, 2017; Wang et al., 2021; Hsu et al., 2022; Tang et al., 2024), while these methods lack theoretical guarantees of quantification. Conformal prediction (CP), on the other hand, offers a systematic approach to construct prediction sets that contain ground-truth labels with a desired coverage guarantee  (Vovk et al., 2005; Romano et al., 2020; Angelopoulos et al., 2021; Huang et al., 2023a; Xi et al., 2024).

CP algorithms utilize non-conformity scores to measure dissimilarity between a new instance and the training instances. The lower the score of a new instance, the more likely it belongs to the same distribution space as the training instances, thereby included in the prediction set. To improve the efficiency of prediction sets for GNNs, DAPS (Zargarbashi et al., 2023) smooths node-wise non-conformity scores by incorporating neighborhood information based on the assumption of network

---

*Corresponding author (`chjwang@nju.edu.cn`)

homophily. Similar to DAPS, CF-GNN (Huang et al., 2023b) introduces a topology-aware output correction model that learns to update prediction and then produces more efficient prediction sets or intervals with the inefficiency as the optimization objective. However, they only consider structural neighbors and ignore the effect of other nodes that are far from the ego node. This motivates us to analyze the influence of global nodes on the size of prediction sets.

In this work, we show that aggregating the information of global nodes with the same label as the ego node benefits the performance of CP methods. We provide an empirical analysis by randomly selecting nodes with the same label as the ego node from an oracle perspective, where the ground-truth labels of all nodes are known, and then aggregating their non-conformity scores into the ego node. The results indicate that aggregating scores of these nodes can significantly reduce the average size of prediction sets. This suggests that the information of nodes with the same label could correct the non-conformity scores, thereby prompting the efficiency of prediction sets. Detailed analysis is presented in Subsection 3.1. However, during the testing phase, the ground-truth label of every test node is unknown. Inspired by the analysis, our key idea is to accurately identify and select as many nodes with the same label as the ego node as possible and aggregate their non-conformity scores.

To this end, we propose a novel algorithm named **S**imilarity-**N**avigated **A**daptive **P**rediction **S**ets (SNAPS), which could self-adaptively aggregate the non-conformity scores of other nodes into the ego node. Specifically, SNAPS gives the higher cumulative weight for nodes with a higher probability of having the same label as the ego node while preserving its own and the one-hop neighbors. We utilize the feature similarity between nodes and the adjacency matrix to calculate the aggregating weights. In this way, the corrected scores could achieve compact prediction sets while maintaining the desired coverage.

To verify the effectiveness of our method, we conduct thorough empirical evaluations on 10 datasets, including both small datasets and large-scale datasets, e.g., OGBN Products (Bhatia et al., 2016). The results demonstrate that SNAPS not only achieves the pre-defined empirical marginal coverage but also achieves better performance over the compared methods. For example, on OGBN Products, our method reduces the average size of prediction sets from 14.92 of APS to 7.68. Moreover, we adapt SNAPS to image classification problems. The results demonstrate that SNAPS reduces the average size of prediction sets from 19.639 to 4.079 – only $\frac{1}{5}$ of the prediction set size from APS on ImageNet (Deng et al., 2009). Code is available at `https://github.com/janqsong/SNAPS`.

We summarize our contributions as follows:

- We empirically explain that non-conformity scores of nodes with the same label as the ego node play a critical role in their non-conformity scores.
- We propose a novel algorithm, namely SNAPS that aggregates basic non-conformity scores of nodes obtained through node feature similarity and one-hop structural neighborhood. We provide theoretical analysis to show the marginal coverage properties of SNAPS and the validity of SNAPS.
- Extensive experimental results demonstrate the effectiveness of our proposed method. We show that SNAPS not only maintains the pre-defined coverage but also achieves great performance in efficiency and singleton hit ratio.

## 2 Preliminary

In this paper, we focus on split conformal prediction for semi-supervised node classification with transductive learning in an undirected graph.

**Notation.** Graph is represented as $\mathcal{G} = (\mathcal{V}, \mathcal{E})$, where $\mathcal{V} := \{v_i\}_{i=1}^N$ denotes the node set and $\mathcal{E}$ denotes the edge set with $|\mathcal{E}| = E$. Let $\boldsymbol{A} \in \{0,1\}^{N \times N}$ be the adjacency matrix, where $\boldsymbol{A}_{i,j} = 1$ if there exists an edge between nodes $v_i$ and $v_j$, and $\boldsymbol{A}_{i,j} = 0$ otherwise, and $\boldsymbol{D}$ be its degree matrix, where $\boldsymbol{D}_{i,i} = \sum_j \boldsymbol{A}_{i,j}$. Let $\boldsymbol{X} := [\boldsymbol{x}_1, \dots, \boldsymbol{x}_N]^T$ be the node feature matrix, where $\boldsymbol{x}_i \in \mathbb{R}^d$ is a $d$-dimensional feature vector for node $v_i$. The label of node $v_i$ is $y_i \in \mathcal{Y}$, where $\mathcal{Y} := \{1, 2, ..., K\}$ denotes the label space.

**Transductive setting.** In transductive setting, we have access to two node sets, $\mathcal{V}_{\text{label}}$ with labels and $\mathcal{V}_{\text{unlabel}}$ without labels, where $\mathcal{V}_{\text{label}} \cap \mathcal{V}_{\text{unlabel}} = \emptyset$ and $\mathcal{V}_{\text{label}} \cup \mathcal{V}_{\text{unlabel}} = \mathcal{V}$. $\mathcal{V}_{\text{label}}$ is then randomly split into $\mathcal{V}_{\text{train}}/\mathcal{V}_{\text{valid}}/\mathcal{V}_{\text{calib}}$ with a fixed size, the training/validation/calibration node set, correspondingly.

$\mathcal{V}_{\text{unlabel}}$ is used as the testing node set $\mathcal{V}_{\text{test}}$. The classifier $f(\cdot)$ is trained on $\{(\boldsymbol{x}_i, y_i)\}_{v_i \in \mathcal{V}_{\text{train}}}$, $\{\boldsymbol{x}_i\}_{v_i \in \mathcal{V} - \mathcal{V}_{\text{train}}}$ and the entire graph structure $\mathcal{G} = (\mathcal{V}, \mathcal{E})$, and is chosen through $\{(\boldsymbol{x}_i, y_i)\}_{v_i \in \mathcal{V}_{\text{valid}}}$. Then we can get the predicted probability $\boldsymbol{P} = \{\boldsymbol{p}_i\}_{v_i \in \mathcal{V}}$ for each node through $\boldsymbol{p}_i = \sigma(f(\boldsymbol{x}_i))$ where $\boldsymbol{p}_i \in [0, 1]^K$ and $\sigma$ is activation function such as softmax. We usually choose the label with the highest probability as the predicted label, i.e., $\hat{y}_i = \arg\max_k \boldsymbol{p}_{ik}$.

**Graph neural networks.** GNNs aim at learning representation vectors for nodes in the graph by leveraging graph structure and node features. Most modern GNNs adopt a series of propagation layers following a message passing mechanism (Gilmer et al., 2017). The $l$-th layer of the GNNs takes the following form:

$$\boldsymbol{h}_i^{(l)} = \text{COMBINE}^{(l)}\left(\boldsymbol{h}_i^{(l-1)}, \text{AGG}^{(l)}\left(\left\{\text{MSG}^{(l)}(\boldsymbol{h}_j^{(l-1)}, \boldsymbol{h}_i^{(l-1)}) | v_j \in \mathcal{N}_i\right\}\right)\right) \quad (1)$$

where $\boldsymbol{h}_i^{(l)}$ is the hidden representation of node $v_i$ at the $l$-th layer with initialization of $\boldsymbol{h}_i^{(0)} = \boldsymbol{x}_i$, and $\mathcal{N}_i$ is a set of nodes adjacent to node $v_i$. $\text{MSG}^{(l)}(\cdot)$, $\text{AGG}^{(l)}(\cdot)$ and $\text{COMBINE}^{(l)}(\cdot)$ denote the functions for message computation, message aggregation, and message combination, respectively. After an iteration of the last layer, the obtained final node representation $\boldsymbol{H} = \{\boldsymbol{h}_i^L\}_{v_i \in \mathcal{V}}$ is then fed to a classifier to obtain the predicted probability $\boldsymbol{P}$.

**Conformal prediction.** CP is a promising framework for generating prediction sets that statistically contain ground-truth labels with a desired guarantee. Formally, given calibration data $\mathcal{D}_{\text{calib}} = \{(\boldsymbol{x}_i, y_i)\}_{i=1}^n$, we can generate a prediction set $\mathcal{C}(\boldsymbol{x}_{n+1}) \subseteq \mathcal{Y}$ for an unseen instance $\boldsymbol{x}_{n+1}$ with the coverage guarantee $\mathbb{P}[y_{n+1} \in \mathcal{C}(\boldsymbol{x}_{n+1})] \geq 1 - \alpha$, where $\alpha$ is the pre-defined significance level. The best characteristic of CP is that it is distribution-free and only relies on exchangeability. This means that every permutation of the instances is equally likely, i.e., $\mathcal{D}_{\text{calib}} \cup (\boldsymbol{x}_{n+1}, y_{n+1})$ is exchangeable, where $(\boldsymbol{x}_{n+1}, y_{n+1})$ is an unseen instance.

Conformal prediction is typically divided into two types: full conformal prediction and split conformal prediction. Unlike full conformal prediction, split conformal prediction treats the model as a black box, avoiding the need to retrain or modify the model and sacrificing efficiency for computational efficiency (Vovk et al., 2005; Zargarbashi et al., 2023). In this paper, we focus on the computationally efficient split conformal prediction method, thus "conformal prediction" in the following denotes split conformal prediction.

**Theorem 1** *(Vovk et al., 2005) Let calibration data and a test instance, i.e., $\{(\boldsymbol{x}_i, y_i)\}_{i=1}^n \cup \{(\boldsymbol{x}_{n+1}, y_{n+1})\}$ be exchangeable. For any non-conformity score function $s : \mathcal{X} \times \mathcal{Y} \to \mathbb{R}$ and any significance level $\alpha \in (0, 1)$, define the $1 - \alpha$ quantile of scores as $\hat{q} := \text{Quantile}\left(\frac{\lceil(1-\alpha)(n+1)\rceil}{n}; \{s(\boldsymbol{x}_i, y_i)\}_{i=1}^n\right)$ and prediction sets as $\mathcal{C}_\alpha(\boldsymbol{x}_{n+1}) = \{y | s(\boldsymbol{x}_{n+1}, y) \leq \hat{q}\}$. We have*

$$1 - \alpha \leq \mathbb{P}[y_{n+1} \in \mathcal{C}_\alpha(\boldsymbol{x}_{n+1})] < 1 - \alpha + \frac{1}{n+1}. \quad (2)$$

Theorem 1 statistically provides a marginal coverage guarantee for all test instances. Currently, there are already many basic non-conformity score methods (Romano et al., 2020; Angelopoulos et al., 2021; Huang et al., 2023a). Here we provide the definition of Adaptive Prediction Sets (Romano et al., 2020) (APS).

**Adaptive Prediction Sets.** In the APS method, the non-conformity scores are calculated by accumulating the softmax probabilities in descending order. Formally, given a data pair $(\boldsymbol{x}, y)$ and a predicted probability estimator $\pi(\boldsymbol{x})_y$ for $(\boldsymbol{x}, y)$, where $\pi(\boldsymbol{x})_y$ is the predicted probability for class $y$, the non-conformity scores can be computed by:

$$s(\boldsymbol{x}, y) = \sum_{i=1}^{|\mathcal{Y}|} \pi(\boldsymbol{x})_i \mathbb{I}[\pi(\boldsymbol{x})_i > \pi(\boldsymbol{x})_y] + \xi \cdot \pi(\boldsymbol{x})_y, \quad (3)$$

where $\xi \in [0, 1]$ is a uniformly distributed random variable. Then, the prediction set is constructed as $\mathcal{C}(\boldsymbol{x}) = \{y | s(\boldsymbol{x}, y) \leq \hat{q}\}$.

**Evaluation Metrics.** The goal is to improve the efficiency of conformal prediction sets as much as possible while maintaining the empirical marginal coverage guarantee. Given the testing nodes set $\mathcal{V}_{\text{test}}$, the efficiency is defined as the average size of prediction sets: $\text{Size} := \frac{1}{|\mathcal{V}_{\text{test}}|} \sum_{v_i \in \mathcal{V}_{\text{test}}} |\mathcal{C}(\boldsymbol{x}_i)|$.

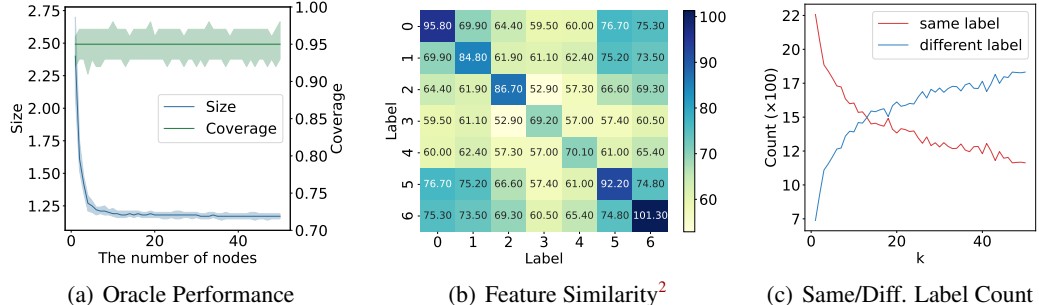

|  | (a) Oracle Performance | (b) Feature Similarity[2] | (c) Same/Diff. Label Count |

Figure 1: The motivation for SNAPS. (a) The trend of Coverage and Size as the number of nodes with the same label as the ego node increases. (b) The average of node feature cosine similarity between same or different labels. (c) The number statistics of nodes with the same label and with different labels as the ego node with increasing $k$ that denotes $k$-NN with feature similarity.

The smaller the size, the more efficient CP is. The empirical marginal coverage is defined as Coverage $:= \frac{1}{|\mathcal{V}_{\text{test}}|} \sum_{v_i \in \mathcal{V}_{\text{test}}} \mathbb{I}[y_i \in \mathcal{C}(\boldsymbol{x}_i)]$. Although efficiency is a common metric for evaluating CP, singleton hit ratio (SH), defined as the proportion of prediction sets of size one that contains the ground-truth label, is also important (Zargarbashi et al., 2023). The formula of SH is defined as: SH $:= \frac{1}{|\mathcal{V}_{\text{test}}|} \sum_{v_i \in \mathcal{V}_{\text{test}}} \mathbb{I}[\mathcal{C}(\boldsymbol{x}_i) = \{y_i\}]$.

## 3 Motivation and Methodology

In this section, we begin by outlining our motivation, substantiating its validity and feasibility through experimental evidence. Then, we propose our method, SNAPS. Finally, we demonstrate that SNAPS satisfies the exchangeability assumption required by CP and offer proof of its improved efficiency compared to basic non-conformity score methods.

### 3.1 Motivation

In this subsection, we empirically show that nodes with the same label as the ego node may play a critical role in the non-conformity scores of the ego node. Specifically, using the scores of nodes with the same label to correct the scores of the ego node could reduce the average size of prediction sets.

To analyze the role of nodes with the same label as the ego node, assuming we have access to an oracle graph, i.e., the ground-truth labels of all the nodes are known. Then, we randomly select nodes with the same label as the ego node and aggregate their APS non-conformity scores into the ego node. We conduct experiments by Graph Convolutional Network (GCN) (Kipf and Welling, 2017) on CoraML (McCallum et al., 2000) dataset and choose APS as the basic score function of CP. Then, we conduct 10 trials and randomly select 100 calibration sets for each trial to evaluate the performance of CP at a significance level $\alpha = 0.05$.

In Figure 1(a), we can find that the average size of prediction sets drops sharply as the number of nodes being aggregated increases, while maintaining valid coverage. When the number of selected nodes is 0, the results shows the performance of APS. Therefore, if the non-conformity scores of the ego node are corrected by accurately selecting nodes with the same label, Size can be reduced to a large extent. Moreover, aggregating the scores of these nodes still achieves the coverage guarantee.

### 3.2 Similarity-Navigated Adaptive Prediction Sets

In our previous analysis, we show that correcting the scores of the ego node with the scores of nodes having the same label leads to smaller prediction sets and valid coverage. However, the above experiment is based on the oracle graph. In the real-world application, the ground-truth label of each test node is unknown. To alleviate this issue, our key idea is to use the similarity to approximate the

---

[2]Values of feature similarity are multiplied by 1000.

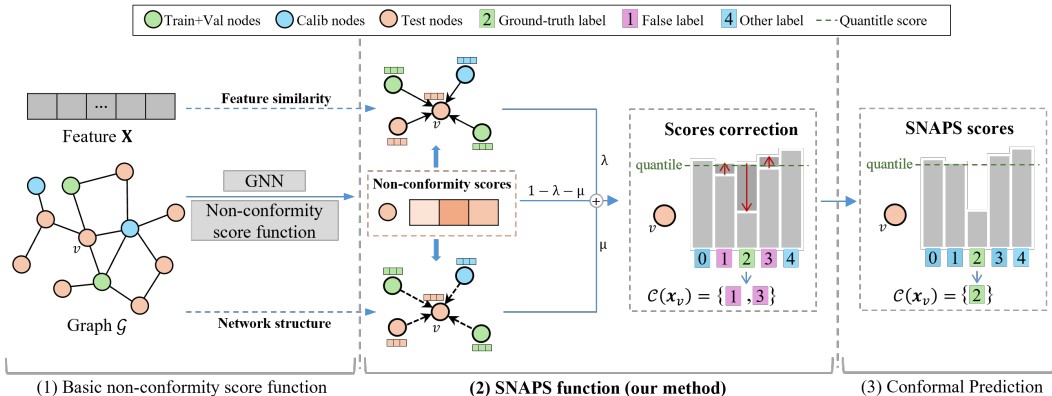

Figure 2: The overall framework of SNAPS. (1) Basic non-conformity score function. We first use basic non-conformity score functions, e.g., APS, to convert node embeddings into non-conformity scores. (2) SNAPS function. We then aggregate basic non-conformity scores of $k$-NN with feature similarity and one-hop structural neighbors to correct the non-conformity scores of nodes. (3) Conformal Prediction. Finally, we use conformal prediction to generate prediction sets, significantly reducing their size compared to the basic score functions.

potential label. Specifically, the nodes with high feature similarity tend to have a high probability of belonging to the same label.

Several studies (Jin et al., 2021b;a; Zou et al., 2023) have demonstrated that matrix constructed from node feature similarity can help with the homophily assumption, i.e., connected nodes in the graph are likely to share the same label. Additionally, the network homophily can also help us to find more nodes whose labels are likely to be the same as the ego node (Kipf and Welling, 2017), and several studies have demonstrated the effectiveness of this (Clarkson, 2023; Zargarbashi et al., 2023; Huang et al., 2023b). Therefore, we consider feature similarity and network structure to select nodes that may have the same label as the ego node.

**Feature similarity graph construction.** We compute the cosine similarity between the node features in the graph. For a given node pair $(v_i, v_j)$, the cosine similarity between their features can be calculated by:

$$\text{Sim}(i, j) = \frac{\boldsymbol{x}_i^\top \boldsymbol{x}_j}{\|\boldsymbol{x}_i\|_2 \cdot \|\boldsymbol{x}_j\|_2}, \tag{4}$$

where $i \neq j$, $v_i \in \mathcal{V}$ and $v_j \in \mathcal{V}_{t,i}$. Here, $v_j \in \mathcal{V}_{t,i}$ represents a set of nodes for which we calculate the similarity with $v_i$. Then, we choose $k$ nearest neighbors for each node based on the above cosine similarity, forming the $k$-NN graph. We denote the adjacency matrix of $k$-NN graph as $\boldsymbol{A}_s$ and its degree matrix as $\boldsymbol{D}_s$, where $\boldsymbol{A}_s(i, j) = \text{Sim}(i, j)$ and $\boldsymbol{D}_s(i, i) = \sum_j \boldsymbol{A}_s(i, j)$. For large graphs, we randomly select $M \gg k$ nodes to put into $\mathcal{V}_{t,i}$, whereas for small graphs, we include all nodes into $v_j \in \mathcal{V}_{t,i}$.

To verify the effectiveness of feature similarity, we provide an empirical analysis. Figure 1(b) presents the average of node feature cosine similarity between the same or different labels on the CoraML dataset. We can find that the average of node feature similarity between the same label is higher than those between different labels. We analyze experimentally where using feature similarity to select $k$-NN meets our expectation of selecting nodes with the same label as the ego node. Figure 1(c) shows the number statistics of nodes with the same label and with different labels at $k$-th nearest neighbors. The result shows that we can indeed select many nodes with the same label when $k$ is not very large.

**SNAPS.** We propose SNAPS that aggregates non-conformity scores of nodes with high feature similarity to ego node and one-hop graph structural neighbors. Formally, for a node $v_i$ with a label $y$, the score function of SNAPS is shown as :

$$\hat{s}(\boldsymbol{x}_i, y) = (1 - \lambda - \mu)s(\boldsymbol{x}_i, y) + \frac{\lambda}{\boldsymbol{D}_s(i, i)} \sum_{j=1}^{M} \boldsymbol{A}_s(i, j)s(\boldsymbol{x}_j, y) + \frac{\mu}{|\mathcal{N}_i|} \sum_{v_j \in \mathcal{N}_i} s(\boldsymbol{x}_j, y), \tag{5}$$

where $s(\cdot, \cdot)$ is the basic non-conformity score function and $\hat{s}(\cdot, \cdot)$ is the SNAPS score function. Both $\lambda$ and $\mu$ are hyperparameters, which are used to measure the importance of three parts of non-conformity scores. The framework of SNAPS is shown in Figure 2 and the pseudo-code is in Appendix B.

## 3.3 Theoretical Analysis

To deploy CP for graph-structured data, the only assumption we should satisfy is exchangeability, i.e., the joint distribution of calibration and testing sets remains unchanged under any permutation. Several studies have demonstrated that non-conformity scores based on the node embeddings obtained by any GNN models are invariant to the permutation of nodes while permuting their edges correspondingly in the calibration and testing sets. This invariance arises because GNNs models and non-conformity score functions only use the structures and attributes in the graph, without dependence on the order of the nodes (Zargarbashi et al., 2023; Huang et al., 2023b). Under this condition, we prove that SNAPS non-conformity scores are still exchangeable.

**Proposition 1** *Let $S = \{s_i\}_{v_i \in \mathcal{V}}$ be basic non-conformity scores of nodes, where $s_i \in \mathbb{R}^K$. Assume that $S$ is exchangeable for all $v_i \in (\mathcal{V}_{calib} \cup \mathcal{V}_{test})$. Then the aggregated $\hat{S} = (1 - \lambda - \mu)S + \lambda \hat{A}_s S + \mu \hat{A} S$, where $\hat{A}_s = D_s^{-1} A_s$ and $\hat{A} = D^{-1} A$, is also exchangeable for $v_i \in (\mathcal{V}_{calib} \cup \mathcal{V}_{test})$.*

The corresponding proof is provided in Appendix A. We then demonstrate the validity of our method theoretically.

**Proposition 2** *Assume that all of the nodes aggregated by SNAPS are the same label as the ego node. Given a data pair $(\boldsymbol{x}, y)$ and a predicted estimator $\pi(\boldsymbol{x})_y$ for $(\boldsymbol{x}, y)$, where $\pi(\boldsymbol{x})_y$ is the predicted probability for class $y$. Moreover, $\epsilon_{ki}$ reflects the model's error in misclassifying the ground-truth label $k$ as label $i$. Let $S$ be APS scores of nodes, where $S_{ui} \in [0, 1]$ is the score corresponding to node $u$ with label $i$. Let $E_k[\pi(\boldsymbol{x}_u)_i]$ and $E_k[S_{ui}]$ be the average of predicted probability and scores corresponding to label $i$ of nodes whose ground-truth labels are $k$, respectively. Let $\eta$ be $1 - \alpha$ quantile of basic non-conformity scores with a significance level $\alpha$. If $E_k[S_{uk}] < \eta$ and $E_k[S_{ui}] \geq (1 - \epsilon_{ki})E_k[\pi(\boldsymbol{x}_u)_{max}] + E_k[\xi \cdot \pi(\boldsymbol{x}_u)_i]$, where $E_k[\pi(\boldsymbol{x}_u)_{max}]$ denotes the maximum predicted probability of nodes whose ground-truth labels are $k$ and $\xi \in [0, 1]$ is a uniformly distributed random variable, then*

$$\mathbb{E}[|\tilde{\mathcal{C}}(\boldsymbol{x})|] \leq \mathbb{E}[|\mathcal{C}(\boldsymbol{x})|],$$

*where $\mathcal{C}(\cdot)$ and $\tilde{\mathcal{C}}(\cdot)$ represent the prediction set from the APS score function and SNAPS score function, respectively.*

In other words, SNAPS consistently generates a smaller prediction set than basic non-conformity scores functions and maintains the desired marginal coverage rate. It is obvious that we can't ignore a very important thing, which is to select nodes with the same label as the ego node as correctly as possible, otherwise, it will lead to a decrease in the efficiency of SNAPS.

## 4 Experiments

In this section, we conduct extensive experiments on semi-supervised node classification to demonstrate the effectiveness of SNAPS on graph-structure data. We also adapt SNAPS for image classification problems. Furthermore, we perform ablation studies and parameter analysis to show the importance of different components in SNAPS and evaluate its robustness, respectively.

### 4.1 Experimental Settings

**Datasets.** In our experiments, we consider ten datasets with high homophily, where connected nodes in the graph are likely to share the same label. These datasets include the common citation graphs: CoraML (McCallum et al., 2000), PubMed (Namata et al., 2012), CiteSeer (Sen et al., 2008), CoraFull (Bojchevski and Günnemann, 2018), Coauthor Physics (Physics) and Coauthor CS (CS) (Shchur et al., 2018) and the co-purchase graphs: Amazon Photos (Photos) and Amazon Computers (Computers) (McAuley et al., 2015; Shchur et al., 2018). Moreover, we consider two large-scale graph datasets, i.e., OGBN Arxiv (Arxiv) (Wang et al., 2020) and OGBN Products (Products) (Bhatia

Table 1: Results of Coverage, Size and SH on different datasets. For SNAPS we use the APS score as the basic score. We report the average calculated from 10 GCN runs with each run of 100 conformal splits at a significance level $\alpha = 0.05$. **Bold** numbers indicate optimal performance.

| Datasets | Coverage | | | | Size↓ | | | | SH%↑ | | | |
|---|---|---|---|---|---|---|---|---|---|---|---|---|
| | APS | RAPS | DAPS | SNAPS | APS | RAPS | DAPS | SNAPS | APS | RAPS | DAPS | SNAPS |
| CoraML | 0.950 | 0.950 | 0.950 | 0.950 | 2.42 | 2.21 | 1.92 | **1.68** | 44.89 | 22.19 | 52.16 | **56.30** |
| PubMed | 0.950 | 0.950 | 0.950 | 0.950 | 1.79 | 1.77 | 1.76 | **1.62** | 33.67 | 30.83 | 35.25 | **42.95** |
| CiteSeer | 0.950 | 0.950 | 0.950 | 0.950 | 2.34 | 2.36 | 1.94 | **1.84** | 50.41 | 38.99 | **59.75** | 59.08 |
| CoraFull | 0.950 | 0.950 | 0.950 | 0.950 | 17.54 | 10.72 | 11.81 | **9.80** | 10.23 | 2.13 | **8.67** | 5.76 |
| CS | 0.950 | 0.950 | 0.950 | 0.950 | 1.91 | 1.20 | 1.22 | **1.08** | 66.17 | 78.34 | 79.80 | **87.92** |
| Physics | 0.950 | 0.950 | 0.950 | 0.950 | 1.28 | 1.07 | 1.08 | **1.04** | 76.74 | 88.89 | 88.40 | **91.21** |
| Computers | 0.950 | 0.950 | 0.950 | 0.950 | 3.95 | 2.89 | 2.13 | **1.98** | 27.67 | 15.85 | 43.03 | **45.48** |
| Photo | 0.951 | 0.950 | 0.950 | 0.951 | 1.89 | 1.64 | 1.41 | **1.31** | 54.31 | 56.63 | 74.57 | **78.51** |
| Arxiv | 0.950 | 0.950 | 0.949 | 0.950 | 4.30 | 3.62 | 3.73 | 3.62 | 22.55 | 14.52 | 19.19 | **23.53** |
| Products | 0.950 | 0.951 | 0.950 | 0.950 | 14.92 | 13.67 | 10.91 | **7.68** | 15.51 | 11.51 | 19.29 | **22.38** |
| Average | 0.950 | 0.950 | 0.950 | 0.950 | 5.23 | 4.12 | 3.79 | **3.17** | 40.22 | 36.00 | 48.01 | **52.31** |

et al., 2016). Particularly, for CoraFull which is highly class-imbalanced, we filter out the classes with fewer than 50 nodes. The transformed dataset is dubbed as CoraFull* (Zargarbashi et al., 2023). Detailed statistics of these datasets are shown in Appendix F. In addition to the datasets mentioned above, we discuss two heterophilous graph datasets in Appendix C.1, namely Chameleon and Squirrel, both of which are two Wikipedia networks (Rozemberczki et al., 2021).

**Baselines.** Since our SNAPS is a general post-processing method for GNNs, here we choose GCN (Kipf and Welling, 2017), GAT (Velickovic et al., 2018) and APPNP (Gasteiger et al., 2018) as structure-aware models and MLP as a structure-independent model. Moreover, our SNAPS can be based on general conformal prediction non-conformity scores, here we choose APS (Romano et al., 2020) and RAPS (Angelopoulos et al., 2021). For comparison, we compare not only with the basic scores, i.e., APS and RAPS, but also with DAPS (Zargarbashi et al., 2023) for GNNs.

**CP Settings.** For the basic model GCN, GAT, APPNP and MLP, we follow parameters suggested by (Zargarbashi et al., 2023). For DAPS, we follow the official implementation. Since GNNs are sensitive to splits, especially in the sparsely labeled setting (Shchur et al., 2018), we train the model over ten trials using varying train/validation splits. For per class in the training/validation set, we randomly select 20 nodes. For Arxiv and Products dataset, we follow the official split in PyTorch Geometric (Fey and Lenssen, 2019). Then, the remaining nodes are included in the calibration set and the test set. The calibration set ratio is suggested by (Huang et al., 2023b), i.e., modifying the calibration set size to $|\mathcal{V}_{\text{calib}}| = \min\{1000, |\mathcal{V}_{\text{calib}} \cup \mathcal{V}_{\text{test}}|/2\}$. For each trained model, we conduct 100 random splits of calibration/test set. Thus, we totally conduct 1000 trials to evaluate the effectiveness of CP. For the non-conformity score function that requires hyper-parameters, we split the calibration set into two sets, one for tuning parameters, and the other for conformal calibration (Zargarbashi et al., 2023). For SNAPS, we choose $\lambda$ and $\mu$ in increments of 0.05 within the range 0 to 1, and ensure that $\lambda + \mu <= 1$. Each experiment is done with a single NVIDIA V100 32GB GPU.

## 4.2 Experimental results

**SNAPS generates smaller prediction sets and achieves a higher singleton hit ratio.** Table 1 shows that Coverage of all conformal prediction methods is close to the desired coverage $1 - \alpha$. At a significance level $\alpha = 0.05$, Size and SH exhibit superior performance. For example, when evaluated on Products, SNAPS reduces Size from 14.92 of APS to 7.68. Overall, the experiments show that SNAPS has the desired coverage rate and gets smaller Size and higher SH than APS, RAPS, and DAPS. Detailed results for other basic models and SNAPS based on RAPS are available in Appendix D.

**SNAPS generates smaller average prediction sets for each label.** We conduct additional experiments to analyze the average performance of APS and SNAPS on nodes belonging to the same label at a significance level $\alpha = 0.05$. Figure 3(a) shows that the distribution of the average non-conformity scores for nodes belonging to the same label aligns with the assumptions made in Proposition 2, i.e., $E_k[\boldsymbol{S}_{uk}] < \eta$ and $E_k[\boldsymbol{S}_{ui}] - \eta \geq -\Delta$, where $\Delta = \eta - (1 - \epsilon_{ki})E_k[\pi(\boldsymbol{x}_u)_{max}] - E_k[\xi \cdot \pi(\boldsymbol{x}_u)_i]$. If $\Delta > 0$, then it is very small. Size of prediction sets corresponding to APS is 3.29. Figure 3(b) shows

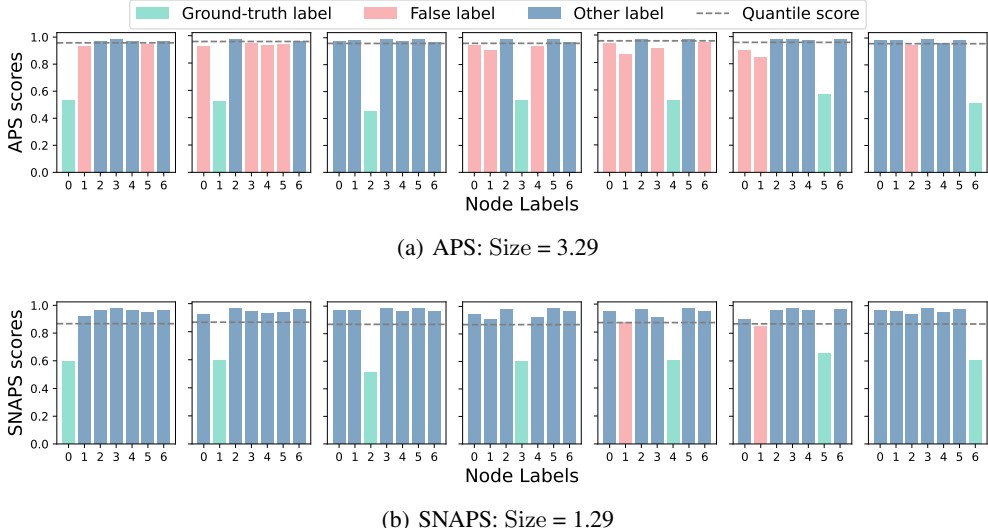

(a) APS: Size = 3.29

(b) SNAPS: Size = 1.29

Figure 3: The average non-conformity scores of nodes belonging to each label based on the model GCN for dataset CoraML.

Table 2: Ablation study in terms of Size. Overall, three parts of our method are critical, and removing any of them results in a general decrease in performance.

| Orig. | Neigh. | Feat. | CoraML | PubMed | CiteSeer | CoraFull* | CS | Physics | Computers | Photo | arxiv | products |
|---|---|---|---|---|---|---|---|---|---|---|---|---|
| ✓ | × | × | 2.42 | 1.79 | 2.34 | 17.54 | 1.91 | 1.28 | 3.95 | 1.89 | 4.30 | 14.92 |
| × | ✓ | × | 2.18 | 1.94 | 2.07 | 17.50 | 1.37 | 1.09 | 2.15 | 1.42 | 4.75 | 11.25 |
| × | × | ✓ | 2.40 | 1.65 | 2.52 | 18.07 | 1.11 | **1.03** | 3.26 | 2.60 | 9.45 | 13.89 |
| ✓ | ✓ | × | 1.87 | 1.72 | 1.91 | 12.10 | 1.22 | 1.07 | 2.22 | 1.37 | 3.76 | 10.81 |
| ✓ | × | ✓ | 1.78 | 1.63 | 1.94 | 11.54 | 1.13 | 1.05 | 2.37 | 1.46 | 3.82 | 8.46 |
| × | ✓ | ✓ | 1.72 | 1.63 | 1.86 | 10.51 | 1.09 | 1.04 | **1.94** | 1.31 | 4.44 | **7.65** |
| ✓ | ✓ | ✓ | **1.68** | **1.62** | **1.84** | **9.80** | **1.08** | 1.04 | 1.98 | **1.31** | **3.62** | 7.68 |

that only a few other labels different from real labels have average scores lower than the quantile of scores. Size of prediction sets corresponding to SNAPS is 1.29. Overall, for basic non-conformity scores that match this distribution of our assumptions, SNAPS can achieve superior performance based on these scores. The results of CiteSeer and Amazon Computers datasets are available in Appendix D.

**Ablation study.** To understand the effects of three parts of our method, i.e., original scores (Orig.), neighborhood scores (Neigh.), and feature similarity node scores (Feat.), we conduct a thorough ablation experiment using GCN at $\alpha = 0.05$. In Table 2, SNAPS performs best on most datasets when all three parts are included. Moreover, for the remaining dataset on which SNAPS exhibits comparable performance, all those better cases contain the Feat. part. Overall, each part plays a critical role in CP for GNNs, and removing any will in general decrease performance.

**Parameter analysis.** We conduct additional experiments to analyze the robustness of SNAPS. We choose GCN as the GNNs model and APS as the basic non-conformity score function.

Figure 4(a) and Figure 4(b) demonstrate that the performance of SNAPS significantly improves as $k$ gradually increases from 0. This improvement occurs because the increasing nodes with the same label are selected to enhance the ego node. Subsequently, as $k$ continues to increase, the performance of SNAPS tends to stabilize. On the other hand, we find that when $k$ is extremely large, it appears that nodes with the same label cannot be selected with high accuracy only by feature similarity. Thus, when $k$ is extremely large, performance will decline slightly. Figure 4(c) and Figure 4(d) show that as the values of parameter $\lambda$ and $\mu$ change, the most areas in the heatmaps of Size and SH display similar colors. Overall, SNAPS is robust to the parameter $k$ and is not sensitive to parameters $\lambda$ and $\mu$. To further explore the sensitivity of $\lambda$ and $\mu$ to the performance of SNAPS, we set $\lambda = \mu = 1/3$, which indicating that three components of SNAPS are equally weighted. The experimental results in Table 3 demonstrate that SNAPS performs well with these default hyperparameters on most datasets.

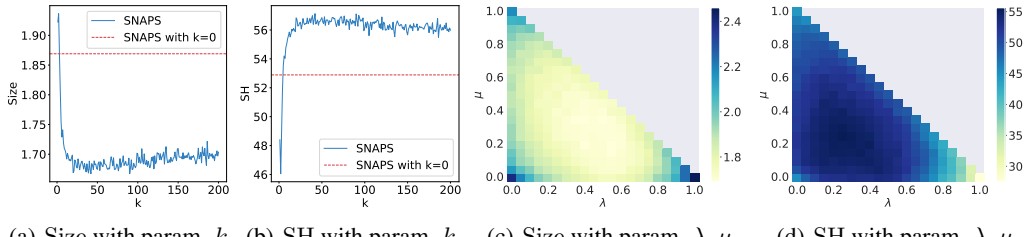

(a) Size with param. $k$  (b) SH with param. $k$  (c) Size with param. $\lambda, \mu$  (d) SH with param. $\lambda, \mu$

Figure 4: Parameter analysis. The results for Size and SH on SNAPS (based on APS) for CoraML dataset with $\alpha = 0.05$.

Table 3: Results of Coverage, Size and SH on different datasets. For SNAPS we use the APS score as the basic score and set $\lambda = \mu = 1/3$. We report the average calculated from 10 GCN runs with each run of 100 conformal splits at a significance level $\alpha = 0.05$. **Bold** numbers indicate optimal performance.

| Datasets | Coverage | | | | Size↓ | | | | SH%↑ | | | |
|---|---|---|---|---|---|---|---|---|---|---|---|---|
| | APS | RAPS | DAPS | SNAPS | APS | RAPS | DAPS | SNAPS | APS | RAPS | DAPS | SNAPS |
| CoraML | 0.950 | 0.958 | 0.957 | 0.951 | 2.50 | 2.62 | 2.32 | **1.74** | 43.09 | 27.34 | 44.52 | **54.11** |
| PubMed | 0.950 | 0.968 | 0.967 | 0.950 | 1.82 | 2.10 | 2.09 | **1.61** | 33.39 | 14.66 | 23.27 | **44.11** |
| CiteSeer | 0.951 | 0.950 | 0.952 | 0.950 | 2.41 | 2.69 | 2.16 | **1.90** | 48.53 | 35.37 | 55.40 | **58.22** |
| CS | 0.950 | 0.953 | 0.954 | 0.950 | 2.04 | 1.31 | 1.33 | **1.13** | 64.32 | 66.91 | 74.91 | **85.21** |
| Physics | 0.951 | 0.962 | 0.962 | 0.950 | 1.39 | 1.44 | 1.28 | **1.07** | 72.44 | 62.22 | 77.65 | **88.58** |
| Computers | 0.950 | 0.950 | 0.951 | 0.950 | 3.01 | 3.04 | 2.30 | **2.01** | 29.21 | 9.87 | 42.19 | **45.98** |
| Photo | 0.949 | 0.950 | 0.950 | 0.950 | 1.90 | 1.81 | 1.56 | **1.30** | 54.86 | 47.27 | 67.57 | **79.50** |

Table 4: Results on Imagenet. The median-of-means is reported over 10 different trials. **Bold** numbers indicate optimal performance.

| | Accuracy | | APS/SNAPS | | | | | |
|---|---|---|---|---|---|---|---|---|
| | | | | $\alpha = 0.1$ | | | $\alpha = 0.05$ | |
| Model | Top1 | Top5 | Coverage | Size ↓ | SSCV ↓ | Coverage | Size↓ | SSCV ↓ |
| ResNeXt101 | 79.32 | 94.58 | 0.899/0.900 | 19.64/**4.08** | 0.088/**0.059** | 0.950/0.950 | 45.80/**14.41** | 0.047/**0.033** |
| ResNet101 | 77.36 | 93.53 | 0.900/0.900 | 10.82/**3.62** | **0.075**/0.078 | 0.950/0.950 | 22.90/**9.83** | 0.039/**0.029** |
| DenseNet161 | 77.19 | 93.56 | 0.900/0.900 | 12.04/**3.80** | 0.077/**0.067** | 0.951/0.950 | 27.99/**10.66** | 0.039/**0.026** |
| ViT | 81.02 | 95.33 | 0.899/0.899 | 10.50/**2.33** | **0.087**/0.133 | 0.949/0.950 | 31.12/**10.47** | 0.042/**0.040** |
| CLIP | 60.53 | 86.15 | 0.899/0.899 | 17.46/**10.32** | 0.047/**0.032** | 0.950/0.949 | 34.93/**24.53** | 0.027/**0.017** |
| Average | - | - | 0.899/0.900 | 14.09/**4.83** | 0.075/**0.074** | 0.950/0.950 | 32.55/**13.98** | 0.039/**0.029** |

**Adaption to image classification problems.** In the node classification problems, SNAPS achieves better performance than standard APS, which was proposed for image classification problems. Therefore, we employ SNAPS for image classification problems. Since there are no links between different images, we utilize the cosine similarities of image features to correct the APS. Formally, the corrected APS, i.e., SNAPS, is defined as :

$$\hat{s}(\boldsymbol{x}, y) = (1 - \eta)s(\boldsymbol{x}, y) + \frac{\eta}{|\mathcal{N}_{\boldsymbol{x}}|} \sum_{\tilde{\boldsymbol{x}} \in \mathcal{N}_{\boldsymbol{x}}} s(\tilde{\boldsymbol{x}}, y),$$

where $s(\boldsymbol{x}, y)$ is the score of standard APS, $\mathcal{N}_{\boldsymbol{x}}$ is the $k$ nearest neighbors based on image features in the calibration set and $\eta$ is a corrected weight. We conduct experiments on ImageNet, whose test dataset is equally divided into the calibration set and the test set. For SNAPS, we set $k = 5$ and $\eta = 0.5$. We report the results of Coverage, Size and *size-stratified coverage violation* (SSCV) (Angelopoulos et al., 2021). The details of experiments and SSCV are provided in Appendix E.

As indicated in Table 4, SNAPS achieves smaller prediction sets than APS. For example, on the ResNeXt101 model and $\alpha = 0.1$, SNAPS reduces Size from 19.639 to 4.079 – only $\frac{1}{5}$ of the prediction set size from APS and achieves the smaller SSCV than APS. Overall, SNAPS could improve the efficiency of prediction sets while maintaining the performance of conditional coverage.

# 5 Related Work

**Uncertainty Quantification for GNNs.** Many uncertainty quantification (UQ) methods have been proposed to quantify the model uncertainty for classification tasks in machine learning (Gal and Ghahramani, 2016; Guo et al., 2017; Zhang et al., 2020; Gupta et al., 2021). Recently, several calibration methods for GNNs have been developed, such as CaGCN (Wang et al., 2021), GATS (Hsu et al., 2022) and SimCalib (Tang et al., 2024). However, these UQ methods lack statistically rigorous and empirically valid coverage guarantee (Huang et al., 2023b). In contrast, SNAPS provides valid coverage guarantees both theoretically and empirically.

**Conformal Prediction for GNNs.** Many conformal prediction (CP) methods have been developed to provide valid uncertainty estimates for model predictions in machine learning classification tasks (Romano et al., 2020; Angelopoulos et al., 2021; Liu et al., 2024; Wei and Huang, 2024). Although several CP methods for GNNs have been studied, the use of CP in graph-structured data is still largely underexplored. ICP (Wijegunawardana et al., 2020) is the first to apply CP framework on graphs, designs a margin conformity score for labels of nodes without considering the relation between nodes. NAPS (Clarkson, 2023) use the non-exchangeable technique from (Barber et al., 2023) for inductive node classification, not applicable for the transductive setting, while we focus on the transductive setting where exchangeability property holds. Our method is essentially an enhanced version of the DAPS (Zargarbashi et al., 2023) method, which proposes a diffusion-based method that incorporates neighborhood information by leveraging the network homophily. Similar to DAPS, CF-GNN (Huang et al., 2023b) introduces a topology-aware output correction model, akin to GCN, which employs a conformal-aware inefficiency loss to refine predictions and improve the efficiency of post-hoc CP. Other recent efforts in CP for graphs include (Lunde, 2023; Marandon, 2023; Zargarbashi and Bojchevski, 2023; Sanchez-Martin et al., 2024) which focus on distinct problem settings. In this work, SNAPS takes into account both network topology and feature similarity. This method can be applied not only to graph-structured data but also to other types of data, such as image data.

# 6 Conclusion

In this paper, we propose SNAPS, a general algorithm that aggregates the non-conformity scores of nodes with the same label as the ego node. Specifically, we select these nodes based on feature similarity and structural neighborhood, and then aggregate their non-conformity scores to the ego node. As a result, our method could correct the scores of some nodes. Moreover, we present theoretical analyses to certify the effectiveness of this method. Extensive experiments demonstrate that SNAPS not only maintains the pre-defined coverage, but also achieves significant performance in efficiency and singleton hit ratio. Furthermore, we extend SNAPS to image classification, where SNAPS shows superior performance compared to APS.

**Limitations.** Our work focuses on node classification using transductive learning. However, in real-world scenarios, many classification tasks require inductive learning. In the future, we aim to apply our method to the inductive setting. Additionally, the method we use to select nodes with the same as the ego node is both computationally inefficient and lacking accuracy. Future work will explore more efficient and accurate methods for node selection. Moreover, while our focus is primarily on datasets with high homophily, many heterophilous networks are prevalent in practice. Consequently, further investigation is essential to enhance the adaptability of SNAPS to these networks.

# Acknowledgments

This paper is supported by the National Natural Science Foundation of China (Grant No. 62192783, 62376117), the National Social Science Fund of China (Grant No. 23BJL035), the Science and Technology Major Project of Nanjing (comprehensive category) (Grant No. 202309007), and the Collaborative Innovation Center of Novel Software Technology and Industrialization at Nanjing University.

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

# A Proofs

In this section, we provided the proofs that were omitted from the main paper.

## A.1 Proof of Proposition 1

*Proof.* (Zargarbashi et al., 2023) have proved that $(1 - \lambda - \mu)\boldsymbol{S} + \mu\hat{\boldsymbol{A}}\boldsymbol{S}$ is exchangeable for $v_i \in (\mathcal{V}_{\text{calib}} \cup \mathcal{V}_{\text{test}})$. So we only need to prove that $\hat{\boldsymbol{A}}_s\boldsymbol{S}$ is also exchangeable for $v_i \in (\mathcal{V}_{\text{calib}} \cup \mathcal{V}_{\text{test}})$. $\hat{\boldsymbol{A}}_s$ is obtained by calculating the feature similarity between two nodes from a global perspective. Before obtaining this matrix, we can not distinguish between labeled and unlabeled nodes, so we just build a new graph structure using node features without considering the order of nodes. So when aggregating non-conformity, we do not break the permutation equivariant. Therefore, $\hat{\boldsymbol{S}} = (1 - \lambda - \mu)\boldsymbol{S} + \lambda\hat{\boldsymbol{A}}_s\boldsymbol{S} + \mu\hat{\boldsymbol{A}}\boldsymbol{S}$ is a special case of a message passing GNNs layer. It follows that $\hat{\boldsymbol{S}}$ is invariant to permutations of the order of the calibration and testing nodes on the graph. Through the proof above, we can conclude that $\hat{\boldsymbol{S}}$ is also exchangeable for $v_i \in (\mathcal{V}_{\text{calib}} \cup \mathcal{V}_{\text{test}})$.

## A.2 Proof of Proposition 2

**Lemma 1** *As stated in Proposition 2, we have*

$$E_k[\boldsymbol{S}_{ui}] \geq (1 - \epsilon_{ki})E_k[\pi(\boldsymbol{x}_u)_{max}] + E_k[\xi \cdot \pi(\boldsymbol{x}_u)_i],$$

*where $E_k[\pi(\boldsymbol{x}_u)_{max}]$ denotes the maximum predicted probability of nodes whose ground-truth labels are $k$, $\epsilon_{ki}$ reflects the model's error in misclassifying the ground-truth label $k$ as label $i$ and $\xi \in [0, 1]$ is a uniformly distributed random variable.*

***Proof of Lemma 1.*** Here, we use APS non-conformity scores as the basic non-conformity scores. Then we have,

$$\boldsymbol{S}_{ui} = \sum_{j=1}^{|\mathcal{Y}|} \pi(\boldsymbol{x}_u)_j \mathbb{I}[\pi(\boldsymbol{x}_u)_j > \pi(\boldsymbol{x}_u)_i] + \xi \cdot \pi(\boldsymbol{x}_u)_i.$$

Suppose $T$ is the number of nodes whose ground-truth label is label $k$. Below we discuss two cases of $\pi(\boldsymbol{x}_u)_i$:

**Case a.** If $\pi(\boldsymbol{x}_u)_i$ is the largest predicted probability for node $u$, then $E_k[\boldsymbol{S}_{ui}] = E_k[\xi \cdot \pi(\boldsymbol{x}_u)_i] = E_k[\pi(\boldsymbol{x}_u)_{max}] + E_k[\xi \cdot \pi(\boldsymbol{x}_u)_i] - E_k[\pi(\boldsymbol{x}_u)_{max}]$. Suppose the number of nodes satisfying this case is $A$.

**Case b.** Otherwise, $E_k[\boldsymbol{S}_{ui}] \geq E_k[\pi(\boldsymbol{x}_u)_{max}] + E_k[\xi \cdot \pi(\boldsymbol{x}_u)_i]$. Suppose the number of nodes satisfying this case is $B$, where $A + B = T$.

Therefore, summing up $E_k[\boldsymbol{S}_{ui}]$ for both cases, we have

$$A \cdot E_k[\boldsymbol{S}_{ui}] + B \cdot E_k[\boldsymbol{S}_{ui}] \geq (A + B) \cdot (E_k[\pi(\boldsymbol{x}_u)_{max}] + E_k[\xi \cdot \pi(\boldsymbol{x}_u)_i]) - A \cdot E_k[\pi(\boldsymbol{x}_u)_{max}].$$

This simplifies to: $E_k[\boldsymbol{S}_{ui}] \geq E_k[\pi(\boldsymbol{x}_u)_{max}] + E_k[\xi \cdot \pi(\boldsymbol{x}_u)_i] - \frac{A}{T} \cdot E_k[\pi(\boldsymbol{x}_u)_{max}]$.

Let $\epsilon_{ki} = \frac{A}{T}$, which reflects the model's error in misclassifying the ground-truth label $k$ as label $i$. Therefore, we conclude that: $E_k[\boldsymbol{S}_{ui}] \geq (1 - \epsilon_{ki})E_k[\pi(\boldsymbol{x}_u)_{max}] + E_k[\xi \cdot \pi(\boldsymbol{x}_u)_i]$.

***Proof of Proposition 2.*** For the sake of description, we denote "$1 - \alpha$ quantile of basic non-conformity scores in the calibrated set" as "the quantile score". Let $\boldsymbol{S}$ and $\hat{\boldsymbol{S}}$ denote APS and SNAPS non-conformity scores, respectively. For node $v$ whose label is $k$, $\hat{\boldsymbol{S}}_v$ can be be expressed as

$$\hat{\boldsymbol{S}}_v = (1 - \lambda)\boldsymbol{S}_v + \frac{\lambda}{|\mathcal{V}_k|} \sum_{u \in \mathcal{V}_k} \boldsymbol{S}_u, \tag{6}$$

where $\mathcal{V}_k$ denotes the nodes set where nodes' ground-truth label is $k$, because regardless of whether high feature similarity nodes or one-hop structural neighbors, the purpose of aggregating these nodes' scores is actually to aggregate, as much as possible, non-conformity scores of nodes with the same label as the ego node.

In order to prove Proposition 2, we only need to prove the following: **1) SNAPS is efficient for the score corresponding to the ground-truth label $k$ of node $v$, i.e., $\hat{S}_{vk} \leq S_{vk}$ or $\hat{S}_{vk} \leq \eta$. 2) SNAPS is efficient for the score corresponding to the other label $i$ of node $v$, i.e., $\hat{S}_{vi} \geq S_{vi}$ or $\hat{S}_{vi} \geq \eta$.** The key idea behind this is as follows. We try to ensure that scores corresponding to the ground-truth label are below the quantile score or decrease compared to the before and scores corresponding to the other label are above the quantile score or increase compared to the before.

**Firstly** SNAPS is efficient for the score corresponding to the ground-truth label $k$ of node $v$, i.e., $\hat{S}_{vk} \leq S_{vk}$ or $\hat{S}_{vk} \leq \eta$. Here we have

$$\hat{S}_{vk} = (1 - \lambda)S_{vk} + \lambda E_k[S_{uk}].$$

1) If $S_{vk} \geq E_k[S_{uk}]$, then

$$\begin{aligned}
\hat{S}_{vk} - S_{vk} &= (1 - \lambda)S_{vk} + \lambda E_k[S_{uk}] - S_{vk} \\
&= -\lambda(S_{vk} - E_k[S_{uk}]) \\
&\leq 0.
\end{aligned}$$

Thus, $\hat{S}_{vk} \leq S_{vk}$. This means that SNAPS can decrease some scores corresponding to the ground-truth label, bringing them from above the quantile score to below it. Since false scores corresponding to ground-truth labels will decrease, $\hat{\eta} < \eta$, where $\hat{\eta}$ denotes $1 - \alpha$ quantile of SNAPS scores in the calibrated set.

2) If $S_{vk} < E_k[S_{uk}]$, then

$$\begin{aligned}
\hat{S}_{vk} &= (1 - \lambda)S_{vk} + \lambda E_k[S_{uk}] \\
&< (1 - \lambda)E_k[S_{uk}] + \lambda E_k[S_{uk}] \\
&= E_k[S_{uk}] \\
&< \eta.
\end{aligned}$$

Thus, $\hat{S}_{vk} < \eta$. This means that for original scores less than the quantile score, they are still less than the quantile score after aggregation.

**Secondly** SNAPS is efficient for the score corresponding to the other label $i$ of node $v$, i.e., $\hat{S}_{vi} \geq S_{vi}$ or $\hat{S}_{vi} \geq \eta$. Here we have

$$\hat{S}_{vi} = (1 - \lambda)S_{vi} + \lambda E_k[S_{ui}].$$

1) If $S_{vi} \leq E_k[S_{ui}]$, then

$$\begin{aligned}
\hat{S}_{vi} - S_{vi} &= (1 - \lambda)S_{vi} + \lambda E_k[S_{ui}] - S_{vi} \\
&= -\lambda(S_{vi} - E_k[S_{ui}]) \\
&\geq 0.
\end{aligned}$$

Thus, $\hat{S}_{vi} \geq S_{vi}$. This means that SNAPS can increase some scores corresponding to the other labels, bringing them from below the quantile score to above it.

2) If $S_{vi} > E_k[S_{ui}]$, then

$$\begin{aligned}
\hat{S}_{vi} - \eta &= (1 - \lambda)S_{vi} + \lambda E_k[S_{ui}] - \eta \\
&> E_k[S_{ui}] - \eta \\
&\geq (1 - \epsilon_{ki})E_k[\pi(x_u)_{max}] + E_k[\xi \cdot \pi(x_u)_i] - \eta.
\end{aligned}$$

Let $\Delta = \eta - (1 - \epsilon_{ki})E_k[\pi(x_u)_{max}] - E_k[\xi \cdot \pi(x_u)_i]$. If $\Delta \leq 0$, then $\hat{S}_{vi} - \eta > 0$. Otherwise, since $\epsilon_{ki}$ reflects the model's error in misclassifying the ground-truth label $k$ as label $i$, and $\hat{\eta} < \eta$, $\Delta$ is very small, which implies that the probability of $\hat{S}_{vi} < \hat{\eta}$ is very low. Therefore, even though some scores corresponding to other false labels may be corrupted, the probability of $\hat{S}_{vi} > \hat{\eta}$ is very high.

**Finally** $\mathbb{E}[|\tilde{\mathcal{C}}(x)|] \leq \mathbb{E}[|\mathcal{C}(x)|]$.

# B Algorithm overview

The pseudo-code for SNAPS is presented in Algorithm 1. First, we preprocess the feature matrix to obtain the similarity graph. Then, we treat the GNNs model as a black box to generate node embeddings. Finally, we perform conformal prediction using SNAPS.

---

**Algorithm 1** Conformal prediction with SNAPS for GNNs pseudo-code

---

**Input:** Graph $\mathcal{G} = \{\mathcal{V}, \mathcal{E}\}$, feature matrix $\boldsymbol{X}$, label set $\mathcal{Y}$, adjacency matrix $\boldsymbol{A}$
  GNNs Model $f$, activation function $\sigma$
  basic score function $s$, SNAPS score function $\hat{s}$
  Calibration data set $\mathcal{D}_{\text{calib}} = \{(\boldsymbol{x}_i, y_i)\}_{i=1}^n$
  test node $v_{n+1}$
  Significance level $\alpha$
1: **Preprocessing**:
  construct similarity graph $\boldsymbol{A}_s$ based on node features
  Train GNNs model $f(\mathcal{G}, \boldsymbol{X})$
2: $\forall v_i \in \mathcal{V}, \boldsymbol{p}_i = \sigma(f(\boldsymbol{x}_i))$
3: $\forall v_i \in \mathcal{V}, \forall k \in \mathcal{Y}$, compute $\boldsymbol{S}_{ik} = s(\boldsymbol{p}_i, k)$
4: $\forall v_i \in \mathcal{V}_{\text{calib}} \cup \{v_{n+1}\}$, compute $\hat{\boldsymbol{S}}_i = \hat{s}(\boldsymbol{S}_i, \boldsymbol{A}_s)$
5: Sort all SNAPS scores $\mathcal{S} = \{\hat{\boldsymbol{S}}_i\}_{i=1}^n$
6: Set $\hat{q} := \text{Quantile}(\frac{\lceil (1-\alpha)(n+1) \rceil}{n}; \mathcal{S})$
7: $\forall k \in \mathcal{V}$, compute $\hat{\boldsymbol{S}}_{n+1,k} = \hat{s}(s(\boldsymbol{x}_{n+1}, k), \boldsymbol{A}_s)$
8: **return** $\mathcal{C}_\alpha(\boldsymbol{x}_{n+1}) = \{k | \hat{\boldsymbol{S}}_{n+1,k} \leq \hat{q}\}$

---

## B.1 Time Complexity

The time complexity of SNAPS is primarily determined by the computation of corrected scores. In this work, we use one-hop nodes and nodes with high feature similarity to correct the ego node. In the transductive setting, this complexity applies to the entire graph. Consequently, one-hop generalization requires $\mathcal{O}(E)$ runtime, while $k$-NN generalization requires $\mathcal{O}(NM)$, where $E$ is the number of edges, $N$ is the number of test nodes, $M$ is the number of nodes sampled to correct the scores of test nodes, with $M \ll N$ for large graphs. For example, we randomly sample $M = 80,000$ nodes from the original set of 2,449,029 nodes in the OGBN Products dataset. Finally, the time complexity of SNAPS is $\mathcal{O}(E + NM)$.

**Time complexity of $k$-NN.** Calculating pairwise similarities is inherently parallelizable, which significantly enhances the efficiency of $k$-NN. Furthermore, there have been some approximation methods that could be used to significantly speed up the computation for large graphs, such as NN-Descent (Dong et al., 2011) that can be easily implemented under MapReduce, achieving an approximate $k$-NN graph in $\mathcal{O}(N^{1.14})$ empirically.

# C Additional Experiments and Experimental Details

## C.1 Experiments on heterophilous graph datasets

To analyze the performance of SNAPS on heterophilous networks, we conduct experiments on two common heterophilous graph datasets. Detailed statistics of two datasets are shown in Appendix F.

We choose FSGNN (Maurya et al., 2022) as the GNN model. We adopt the dataset splits of Geom-GCN (Pei et al., 2020), i.e. splitting nodes into $60\%/20\%/20\%$ for training/validation/testing. To evaluate the performance of the CP methods, we divide the test set equally into the calibration and evaluation sets. For DAPS, we set $\lambda = 0.5$. For SNAPS, we set $\lambda = 0.5, \mu = 0$, and $k = 20$, i.e., neglecting the structural neighborhood. To construct a cosine similarity-based $k$-NN graph for two heterophilous datasets, we utilize the embeddings from FSGNN. In Table 5, empirical results at $\alpha = 0.1$ and $0.15$ are presented. The experimental results show that SNAPS consistently outperforms the baselines on heterophilous networks.

Table 5: Results of Coverage, Size on two heterophilous graph datasets. We report the average calculated from FSGNN with 100 conformal splits at different significance levels ($\alpha = 0.05, 0.1$). **Bold** numbers indicate optimal performance.

| Datasets | Accuracy | $\alpha = 0.1$ | | | | | | $\alpha = 0.15$ | | | | | |
|---|---|---|---|---|---|---|---|---|---|---|---|---|---|
| | | Coverage | | | Size↓ | | | Coverage | | | Size↓ | | |
| | | APS | DAPS | SNAPS | APS | DAPS | SNAPS | APS | DAPS | SNAPS | APS | DAPS | SNAPS |
| Chameleon | 78.09±0.93 | 0.904 | 0.905 | 0.904 | 1.95 | 2.75 | **1.70** | 0.850 | 0.853 | 0.851 | 1.62 | 2.23 | **1.32** |
| Squirrel | 73.72±2.19 | 0.900 | 0.897 | 0.900 | 2.41 | 3.05 | **2.27** | 0.851 | 0.848 | 0.850 | 1.89 | 2.37 | **1.64** |

**Discussion on heterophilous graph datasets.** In this work, we utilize the embeddings from FSGNN to construct the similarity graph. Although this serves as a simple workaround for heterophilous graphs, it does not effectively address the core issue in heterophily, where the expectation is that the embeddings would effectively capture the heterophilous networks. Therefore, we plan to construct the similarity graph based solely on graph structure and node features to enhance the performance of SNAPS in future work.

## C.2 Comparison with CF-GNN

To compare with CF-GNN, we randomly select 20 nodes per class for training/validation, setting the calibration set size to 1,000 (Huang et al., 2023b). The APS scores are used as the basic non-conformity scores. The average results are calculated from 10 GCN with each trial of 100 conformal splits. Moreover, we introduce a new metric, i.e., Time, to evaluate the running time for each trial. As shown in Table 6, SNAPS outperforms CF-GNN in both metrics.

Table 6: Results of Size, Time on three graph datasets for CF-GNN and SNAPS. We report the results at different significance levels ($\alpha = 0.05, 0.1$). **Bold** numbers indicate optimal performance.

| Datasets | Size at $\alpha = 0.05$ | | Size at $\alpha = 0.1$ | | Time | |
|---|---|---|---|---|---|---|
| | CF-GNN | SNAPS | CF-GNN | SNPAS | CF-GNN | SNAPS |
| CoraML | 2.60 | **1.68** | 1.68 | **1.31** | 142s | **0.73s** |
| PubMed | 2.13 | **1.62** | 1.86 | **1.35** | 148s | **0.98s** |
| CiteSeer | 3.07 | **1.84** | 1.96 | **1.39** | 124s | **0.71s** |

# D Detailed Results

In this section, we report the detailed experimental results. Firstly, we report the results of Coverage, Size, SH across different datasets using different GNNs models in Table 7, Table 8, Table 9, Table 10. Then, we report the results of Coverage, Size, SH using RAPS as the basic non-conformity scores in Table 11. Finally, we added two additional datasets CiteSeer and Amazon Computers to observe changes in the distribution of mean scores for each label in Figure 5 and Figure 6. The results show that SNAPS achieves superior performance.

# E Experiments on Image Classification

To verify that our method could be adapted to image classification problems, we test SNAPS on ImageNet and various models. We split the test dataset containing 50000 images into 25000 images for the calibration set and 25000 images for the test set. The models are pretrained Imagenet classifiers from the torchvision repository (Paszke et al., 2019) with standard normalization, resize, and crop parameters. Moreover, all models are calibrated by the Temperature scaling procedure (Guo et al., 2017) on the calibration set. All experiments are conducted with ten trials, and the mean results are reported.

We use *size-stratified coverage violation (SSCV)* (Angelopoulos et al., 2021) to measure how prediction sets violate the conditional coverage. Specifically, considering a disjoint set-size strata $\{S_i\}_{i=1}^{N_s}$, where $\bigcup_{i=1}^{N_s} S_i = \{1, 2, \cdots, |\mathcal{Y}|\}$. Then, we define the indexes of examples stratified by

Table 7: Results of Coverage, Size, SH on different datasets. For SNAPS we use the APS score as the basic score. We report the average calculated from 10 GCN trials with each trial of 100 conformal splits at a significance level $\alpha = 0.1$. **Bold** numbers indicate optimal performance.

| Datasets | Coverage | | | | Size↓ | | | | SH%↑ | | | |
|---|---|---|---|---|---|---|---|---|---|---|---|---|
| | APS | RAPS | DAPS | SNAPS | APS | RAPS | DAPS | SNAPS | APS | RAPS | DAPS | SNAPS |
| CoraML | 0.900 | 0.901 | 0.900 | 0.900 | 1.81 | 1.43 | 1.41 | **1.31** | 54.96 | 56.87 | 65.21 | **69.10** |
| PubMed | 0.900 | 0.900 | 0.900 | 0.900 | 1.48 | 1.43 | 1.44 | **1.35** | 49.61 | 51.36 | 52.58 | **59.14** |
| CiteSeer | 0.900 | 0.900 | 0.900 | 0.900 | 1.85 | 1.46 | 1.43 | **1.39** | 57.86 | 57.17 | 70.11 | **72.56** |
| CoraFull | 0.900 | 0.900 | 0.900 | 0.900 | 10.70 | 5.50 | 6.63 | **5.48** | 16.31 | 1.87 | **18.04** | 15.92 |
| CS | 0.900 | 0.900 | 0.900 | 0.899 | 1.48 | **1.00** | 1.09 | 1.02 | 71.31 | **89.72** | 83.69 | 88.29 |
| Physics | 0.900 | 0.900 | 0.900 | 0.900 | 1.15 | **1.00** | 1.03 | 1.01 | 78.98 | **89.92** | 87.17 | 88.82 |
| Computers | 0.900 | 0.901 | 0.900 | 0.900 | 2.18 | 1.66 | 1.53 | **1.52** | 41.30 | 41.17 | **64.03** | 61.81 |
| Photo | 0.901 | 0.900 | 0.900 | 0.900 | 1.48 | **1.04** | 1.17 | 1.13 | 64.20 | **86.93** | 84.04 | 82.01 |
| Arxiv | 0.899 | 0.900 | 0.899 | 0.900 | 2.81 | **2.12** | 2.66 | 2.30 | 32.83 | 17.00 | **37.60** | 35.24 |
| Products | 0.900 | 0.900 | 0.900 | 0.900 | 6.49 | 3.91 | 3.62 | **3.27** | 34.03 | 21.26 | **48.90** | 44.64 |
| Average | 0.900 | 0.900 | 0.900 | 0.900 | 3.143 | 2.055 | 2.201 | **1.978** | 50.139 | 51.327 | 61.137 | **61.753** |

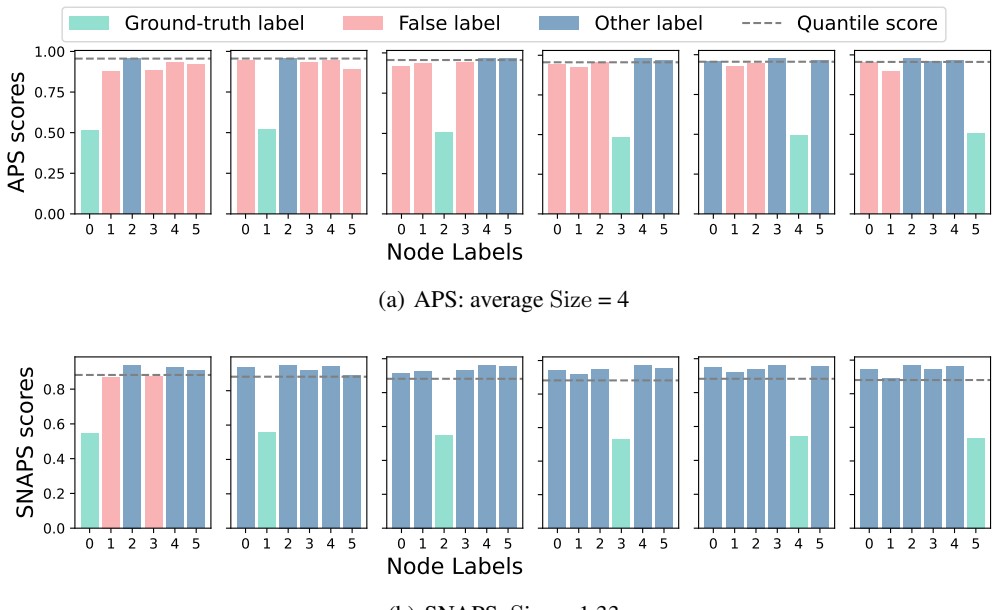

(a) APS: average Size = 4

(b) SNAPS: Size = 1.33

Figure 5: The average non-conformity scores of nodes belonging to each label based on the model GCN for dataset CiteSeer.

the prediction set size by $\mathcal{J}_j = \{i : |\mathcal{C}(\boldsymbol{x}_i)| \in S_j\}$. Formally, we can define the *SSCV* as:

$$\text{SSCV} = \sup_j \left| \frac{|\{i \in \mathcal{J}_j : y_i \in \mathcal{C}(\boldsymbol{x}_i)\}|}{|\mathcal{J}_j|} - (1 - \alpha) \right| \tag{7}$$

where $\alpha$ is the significance level. In our experiments, we follow the setting of RAPS (Angelopoulos et al., 2021), choosing a relatively coarse partitioning of the possible set sizes: 0-1, 2-3, 410, 11-100, and 101-1000.

## F  More Details on Dataset and Models

Table 12 displays the statistics of the dataset used for the evaluation. For CoraFull* we remove the classes (and the respective nodes) that have a number of instances less than 50 in order to have the same number of nodes per class in each train/validation split (Zargarbashi et al., 2023). Table 13 summarizes the model's accuracy on every dataset.

Table 8: Results of Coverage, Size, SH on different datasets. For SNAPS we use the APS score as the basic score. We report the average calculated from 10 GAT trials with each trial of 100 conformal splits at different significance levels ($\alpha = 0.05, 0.1$). **Bold** numbers indicate optimal performance.

| Datasets | Coverage | | | | Size↓ | | | | SH%↑ | | | |
|---|---|---|---|---|---|---|---|---|---|---|---|---|
| | APS | RAPS | DAPS | SNAPS | APS | RAPS | DAPS | SNAPS | APS | RAPS | DAPS | SNAPS |
| | | | | | | | $\alpha = 0.05$ | | | | | |
| CoraML | 0.950 | 0.950 | 0.950 | 0.950 | 2.91 | 2.75 | 2.31 | **1.92** | 30.82 | 15.54 | 41.45 | **46.51** |
| PubMed | 0.950 | 0.950 | 0.950 | 0.950 | 1.83 | 1.82 | 1.79 | **1.70** | 30.00 | 23.83 | 32.55 | **37.22** |
| CiteSeer | 0.950 | 0.950 | 0.950 | 0.950 | 2.61 | 2.58 | 2.18 | **2.08** | 38.82 | 32.87 | 49.43 | **51.25** |
| CoraFull | 0.950 | 0.950 | 0.950 | 0.951 | 25.38 | 17.13 | 17.43 | **13.80** | 2.85 | 0.43 | 2.77 | 1.97 |
| CS | 0.950 | 0.950 | 0.950 | 0.950 | 2.62 | 1.59 | 1.50 | **1.14** | 46.99 | 44.22 | 67.28 | **83.29** |
| Physics | 0.950 | 0.950 | 0.950 | 0.950 | 1.53 | 1.24 | 1.13 | **1.05** | 62.54 | 75.56 | 83.82 | **90.40** |
| Computers | 0.950 | 0.950 | 0.950 | 0.950 | 3.18 | 3.08 | 2.30 | **2.10** | 24.56 | 13.65 | **40.05** | 39.51 |
| Photo | 0.950 | 0.950 | 0.950 | 0.950 | 1.89 | 1.59 | 1.42 | **1.30** | 52.10 | 50.83 | 72.02 | **75.43** |
| Average | 0.950 | 0.950 | 0.950 | 0.950 | 5.244 | 3.973 | 3.758 | **3.136** | 36.085 | 32.116 | 48.671 | **53.198** |
| | | | | | | | $\alpha = 0.1$ | | | | | |
| CoraML | 0.900 | 0.900 | 0.900 | 0.900 | 2.18 | 1.82 | 1.66 | **1.47** | 42.71 | 37.79 | 55.74 | **61.05** |
| PubMed | 0.900 | 0.900 | 0.900 | 0.901 | 1.54 | 1.49 | 1.48 | **1.40** | 44.96 | 45.39 | 49.59 | **54.71** |
| CiteSeer | 0.900 | 0.900 | 0.900 | 0.901 | 2.05 | 1.59 | 1.63 | **1.56** | 49.10 | 50.44 | 61.08 | **63.85** |
| CoraFull | 0.900 | 0.900 | 0.900 | 0.900 | 15.71 | 9.63 | 10.80 | **8.81** | 5.84 | 0.83 | **6.12** | 4.96 |
| CS | 0.900 | 0.900 | 0.899 | 0.900 | 1.90 | 1.10 | 1.21 | **1.04** | 57.68 | 81.9 | 76.72 | **86.89** |
| Physics | 0.900 | 0.900 | 0.901 | 0.900 | 1.30 | **1.01** | 1.05 | 1.02 | 69.42 | **88.92** | 85.74 | 88.39 |
| Computers | 0.900 | 0.900 | 0.900 | 0.900 | 2.15 | 1.82 | 1.66 | **1.60** | 41.43 | 42.13 | **58.05** | 56.54 |
| Photo | 0.900 | 0.900 | 0.900 | 0.901 | 1.48 | **1.07** | 1.21 | 1.15 | 63.36 | **84.31** | 80.94 | 80.16 |
| Average | 0.900 | 0.900 | 0.900 | 0.900 | 3.539 | 2.441 | 2.588 | **2.256** | 46.813 | 53.964 | 59.248 | **62.069** |

Table 9: Results of Coverage, Size, SH on different datasets. For SNAPS we use the APS score as the basic score. We report the average calculated from 10 APPNP trials with each trial of 100 conformal splits at different significance levels ($\alpha = 0.05, 0.1$). **Bold** numbers indicate optimal performance.

| Datasets | Coverage | | | | Size↓ | | | | SH%↑ | | | |
|---|---|---|---|---|---|---|---|---|---|---|---|---|
| | APS | RAPS | DAPS | SNAPS | APS | RAPS | DAPS | SNAPS | APS | RAPS | DAPS | SNAPS |
| | | | | | | | $\alpha = 0.05$ | | | | | |
| CoraML | 0.950 | 0.950 | 0.950 | 0.950 | 2.20 | 2.01 | 1.78 | **1.66** | 46.85 | 27.70 | 55.12 | **54.74** |
| PubMed | 0.950 | 0.950 | 0.950 | 0.950 | 1.75 | 1.75 | 1.74 | **1.67** | 36.42 | 32.58 | 36.79 | **41.25** |
| CiteSeer | 0.950 | 0.950 | 0.950 | 0.950 | 2.18 | 2.22 | 1.88 | **1.86** | 54.08 | 34.40 | **61.85** | 59.74 |
| CoraFull | 0.950 | 0.950 | 0.950 | 0.950 | 14.91 | 9.83 | 11.90 | **9.78** | **10.62** | 2.36 | 8.15 | 4.13 |
| CS | 0.950 | 0.950 | 0.951 | 0.950 | 1.64 | 1.16 | 1.21 | **1.08** | 70.22 | 80.86 | 79.91 | **88.02** |
| Physics | 0.950 | 0.950 | 0.950 | 0.950 | 1.24 | 1.05 | 1.07 | **1.03** | 78.71 | 90.86 | 89.24 | **92.38** |
| Computers | 0.950 | 0.950 | 0.950 | 0.950 | 2.90 | 2.81 | 2.08 | **2.01** | 31.85 | 20.80 | **46.41** | 42.91 |
| Photo | 0.950 | 0.950 | 0.950 | 0.950 | 1.69 | 1.42 | 1.38 | **1.31** | 60.31 | 67.94 | 75.01 | **76.13** |
| Average | 0.950 | 0.950 | 0.950 | 0.950 | 3.564 | 2.781 | 2.880 | **2.550** | 48.633 | 44.688 | 56.560 | **57.413** |
| | | | | | | | $\alpha = 0.1$ | | | | | |
| CoraML | 0.900 | 0.900 | 0.900 | 0.901 | 1.73 | 1.37 | 1.37 | **1.30** | 55.15 | 58.92 | 66.84 | 69.33 |
| PubMed | 0.900 | 0.900 | 0.900 | 0.900 | 1.44 | 1.41 | 1.41 | **1.34** | 53.34 | 52.90 | 55.80 | **59.92** |
| CiteSeer | 0.900 | 0.900 | 0.900 | 0.900 | 1.70 | **1.28** | 1.39 | 1.38 | 62.50 | 69.73 | 72.18 | **72.71** |
| CoraFull | 0.900 | 0.900 | 0.900 | 0.900 | 8.39 | 5.21 | 6.03 | **5.05** | 18.43 | 3.43 | **16.89** | 14.44 |
| CS | 0.901 | 0.900 | 0.901 | 0.900 | 1.34 | **1.00** | 1.08 | 1.01 | 74.42 | **89.70** | 84.28 | 88.75 |
| Physics | 0.900 | 0.900 | 0.900 | 0.900 | 1.13 | **1.00** | 1.02 | 1.01 | 80.36 | **90.00** | 88.05 | 89.42 |
| Computers | 0.900 | 0.900 | 0.900 | 0.900 | 1.96 | 1.56 | 1.50 | **1.48** | 47.78 | 50.06 | **65.94** | 63.93 |
| Photo | 0.900 | 0.900 | 0.900 | 0.900 | 1.39 | **1.06** | 1.19 | 1.14 | 67.47 | **85.46** | 82.26 | 81.42 |
| Average | 0.900 | 0.900 | 0.900 | 0.900 | 2.385 | 1.736 | 1.874 | **1.714** | 57.431 | 62.525 | 66.530 | **67.490** |

Table 10: Results of Coverage, Size, SH on different datasets. For SNAPS we use the APS score as the basic score. We report the average calculated from 10 MLP trials with each trial of 100 conformal splits at different significance levels ($\alpha = 0.05, 0.1$). **Bold** numbers indicate optimal performance.

| Datasets | Coverage | | | | Size↓ | | | | SH%↑ | | | |
|---|---|---|---|---|---|---|---|---|---|---|---|---|
| | APS | RAPS | DAPS | SNAPS | APS | RAPS | DAPS | SNAPS | APS | RAPS | DAPS | SNAPS |
| | | | | | | | $\alpha = 0.05$ | | | | | |
| CoraML | 0.950 | 0.950 | 0.950 | 0.950 | 4.21 | 4.22 | 3.04 | **2.68** | 6.85 | 6.16 | 16.63 | **22.28** |
| PubMed | 0.950 | 0.950 | 0.950 | 0.950 | 2.09 | 2.06 | 2.01 | **1.85** | 18.94 | 7.91 | 21.02 | **27.57** |
| CiteSeer | 0.950 | 0.950 | 0.950 | 0.950 | 4.08 | 4.13 | 3.33 | **3.18** | 11.20 | 10.20 | 17.75 | **21.81** |
| CoraFull | 0.950 | 0.950 | 0.950 | 0.950 | 25.36 | 26.40 | **16.36** | 16.44 | 0.00 | 0.00 | 0.00 | **0.00** |
| CS | 0.950 | 0.950 | 0.950 | 0.950 | 4.36 | 1.57 | 1.69 | **1.31** | 32.81 | 42.01 | 61.93 | **74.87** |
| Physics | 0.950 | 0.950 | 0.950 | 0.950 | 2.00 | 1.62 | 1.19 | **1.12** | 44.98 | 45.63 | 80.08 | **84.63** |
| Computers | 0.950 | 0.950 | 0.950 | 0.950 | 5.35 | 5.34 | **3.97** | 3.99 | 2.35 | 2.14 | **4.79** | 4.48 |
| Photo | 0.950 | 0.950 | 0.950 | 0.950 | 3.20 | 3.12 | **2.12** | 2.16 | 18.14 | 10.66 | **38.22** | 36.91 |
| Average | 0.950 | 0.950 | 0.950 | 0.950 | 6.331 | 6.058 | 4.214 | **4.091** | 16.909 | 15.589 | 30.053 | **34.069** |
| | | | | | | | $\alpha = 0.1$ | | | | | |
| CoraML | 0.900 | 0.901 | 0.900 | 0.900 | 3.20 | 3.23 | 2.17 | **1.93** | 14.10 | 11.16 | 31.24 | **37.95** |
| PubMed | 0.900 | 0.900 | 0.901 | 0.900 | 1.77 | 1.71 | 1.69 | **1.53** | 31.03 | 26.82 | 35.10 | **43.78** |
| CiteSeer | 0.900 | 0.901 | 0.900 | 0.900 | 3.28 | 3.30 | 2.55 | **2.33** | 21.22 | 19.07 | 28.64 | **34.40** |
| CoraFull | 0.900 | 0.900 | 0.900 | 0.900 | 16.34 | 16.35 | **10.27** | 10.31 | 0.00 | 0.00 | 0.00 | 0.00 |
| CS | 0.900 | 0.900 | 0.900 | 0.900 | 3.08 | 1.13 | 1.32 | **1.10** | 40.97 | 78.73 | 71.64 | **82.36** |
| Physics | 0.900 | 0.901 | 0.900 | 0.900 | 1.62 | 1.19 | 1.07 | **1.03** | 54.04 | 75.65 | 84.02 | **87.12** |
| Computers | 0.901 | 0.900 | 0.900 | 0.901 | 3.90 | 3.77 | **2.75** | 2.78 | 7.27 | 5.09 | **17.09** | 16.75 |
| Photo | 0.900 | 0.901 | 0.900 | 0.900 | 2.46 | 2.02 | **1.60** | 1.61 | 27.68 | 19.72 | **54.38** | 53.70 |
| Average | 0.900 | 0.901 | 0.900 | 0.900 | 4.456 | 4.088 | 2.928 | **2.828** | 24.539 | 29.530 | 40.264 | **44.508** |

Table 11: Results of Coverage, Size, SH on different datasets. For SNAPS we use the RAPS score as the basic score. We report the average calculated from 10 GCN trials with each trial of 100 conformal splits at different significance levels ($\alpha = 0.05, 0.1$). **Bold** numbers indicate optimal performance.

| | $\alpha = 0.1$ | | | | | | $\alpha = 0.05$ | | | | | |
|---|---|---|---|---|---|---|---|---|---|---|---|---|
| Datasets | Coverage | | Size↓ | | SH%↑ | | Coverage | | Size↓ | | SH%↑ | |
| | RAPS | SNAPS | RAPS | SNAPS | RAPS | SNAPS | RAPS | SNAPS | RAPS | SNAPS | RAPS | SNAPS |
| CoraML | 0.901 | 0.900 | 1.43 | **1.34** | 56.87 | **61.43** | 0.950 | 0.950 | 2.21 | **1.94** | 22.19 | **33.05** |
| PubMed | 0.900 | 0.900 | 1.43 | **1.38** | 51.36 | **55.29** | 0.950 | 0.950 | 1.77 | **1.64** | 30.83 | **40.39** |
| CiteSeer | 0.900 | 0.900 | 1.46 | **1.36** | 57.17 | **64.44** | 0.950 | 0.950 | 2.36 | **2.02** | 38.99 | **48.14** |
| CoraFull | 0.900 | 0.900 | 5.50 | **5.48** | **1.87** | 1.39 | 0.950 | 0.950 | 10.72 | **10.70** | **2.13** | 0.99 |
| CS | 0.900 | 0.900 | 1.00 | 1.00 | 89.72 | **89.83** | 0.950 | 0.950 | 1.20 | **1.10** | 78.34 | **85.05** |
| Physics | 0.900 | 0.900 | 1.00 | 1.00 | 89.92 | **90.00** | 0.950 | 0.949 | 1.07 | **1.04** | 88.89 | **91.47** |
| Computers | 0.901 | 0.901 | 1.66 | **1.53** | 41.17 | **48.42** | 0.950 | 0.950 | 2.89 | **2.22** | 15.85 | **28.61** |
| Photo | 0.900 | 0.900 | 1.04 | **1.03** | 86.93 | **86.97** | 0.950 | 0.949 | 1.64 | **1.63** | 56.63 | 51.34 |
| Arxiv | 0.900 | 0.901 | 2.12 | 2.17 | **17.00** | 13.85 | 0.950 | 0.950 | 3.62 | **3.61** | 14.52 | **14.59** |
| Products | 0.900 | 0.901 | 3.91 | **3.48** | **21.26** | 21.04 | 0.951 | 0.951 | 13.67 | **7.77** | 11.51 | **23.30** |
| Average | 0.900 | 0.900 | 2.055 | **1.977** | 51.327 | **53.266** | 0.950 | 0.950 | 4.115 | **3.367** | 35.988 | **41.693** |

Table 12: Statistics of the ten datasets.

| Datasets | Nodes | Features | Edges | Classes | Homophily |
|---|---|---|---|---|---|
| CoraML | 2995 | 2879 | 16316 | 7 | 78.85% |
| PubMed | 19717 | 500 | 88648 | 3 | 80.23% |
| CiteSeer | 4230 | 602 | 10674 | 6 | 94.94% |
| CoraFull* | 19749 | 8710 | 126524 | 68 | 56.76% |
| Coauthor CS | 18333 | 6805 | 163788 | 15 | 80.80% |
| Coauthor Physics | 34493 | 8415 | 495924 | 5 | 93.14% |
| Amazon Computers | 13752 | 767 | 491722 | 10 | 77.72% |
| Amazon Photo | 7650 | 745 | 238162 | 8 | 82.72% |
| OGBN Arxiv | 169343 | 128 | 1166243 | 40 | 65.51% |
| OGBN Products | 2449029 | 100 | 123718280 | 49 | 80.75% |
| Chameleon | 2277 | 2325 | 36101 | 5 | 23% |
| Squirrel | 5201 | 2089 | 217073 | 5 | 22% |

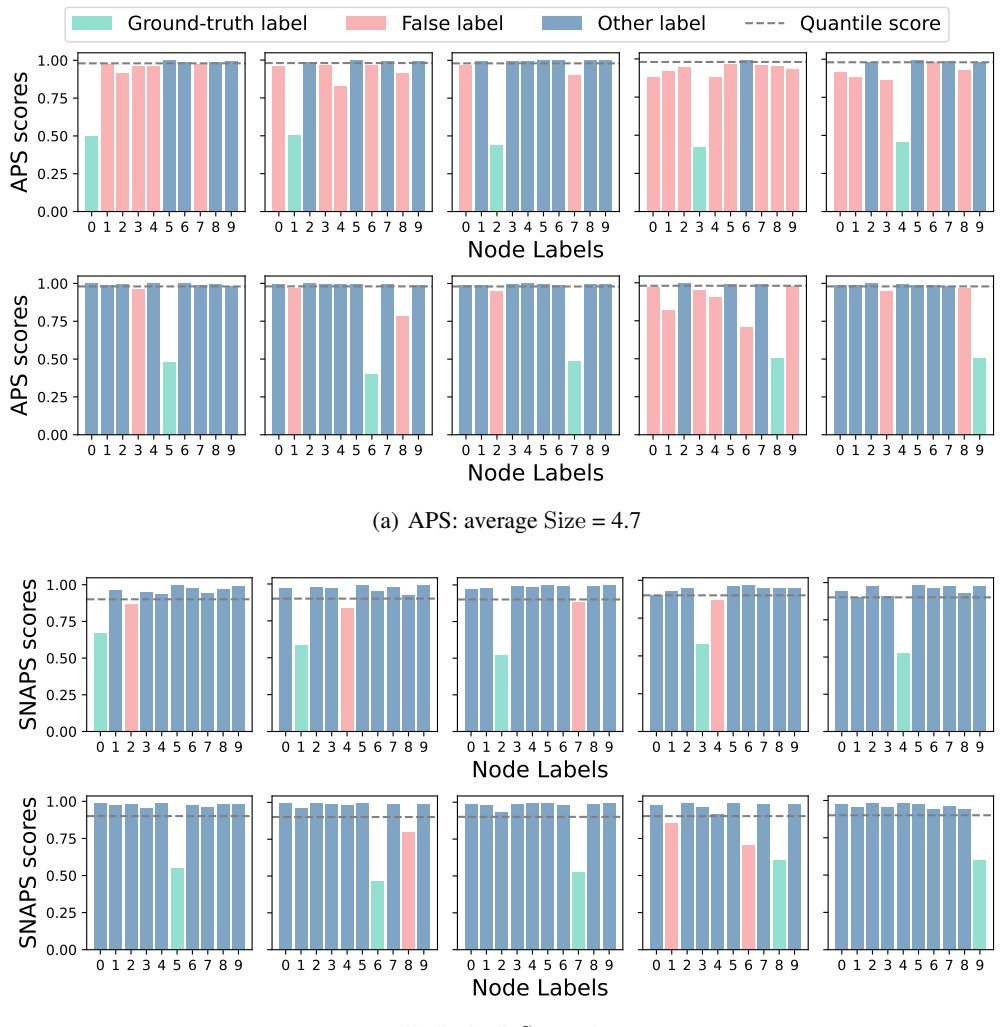

(a) APS: average Size = 4.7

(b) SNAPS: Size = 1.7

Figure 6: The average non-conformity scores of nodes belonging to each label based on the model GCN for dataset Amazon Computers.

Table 13: Accuracy report for datasets and models involved in the analysis.

| Dataset | GCN | GAT | APPNP | MLP |
|---|---|---|---|---|
| CoraML | 81.48±1.80 | 79.09±2.44 | 82.18±1.52 | 62.82±1.85 |
| PubMed | 77.40±1.89 | 75.61±2.75 | 77.97±2.12 | 69.56±1.62 |
| CiteSeer | 83.90±1.04 | 83.03±0.80 | 84.91±0.74 | 62.67±1.11 |
| CoraFull | 60.53±0.62 | 50.19±0.74 | 61.08±0.48 | 38.06±0.65 |
| CS | 91.24±0.40 | 88.01±1.04 | 91.40±0.46 | 88.06±0.57 |
| Physics | 93.11±0.60 | 91.12±1.52 | 93.66±0.48 | 86.93±1.20 |
| Computers | 81.05±1.70 | 77.78±2.29 | 81.36±1.85 | 59.49±3.73 |
| Photo | 89.64±1.21 | 88.49±0.90 | 89.69±2.19 | 74.41±3.68 |
| Arxiv | 70.53 | - | - | - |
| Products | 75.44 | - | - | - |

