# OpenReview forum: "Similarity-Navigated Conformal Prediction for Graph Neural Networks"
_NeurIPS.cc/2024/Conference — NeurIPS 2024 poster_

### Official Review · Reviewer_xCn2 · 2024-07-11

**Soundness:** 3
**Presentation:** 3
**Contribution:** 2
**Rating:** 6
**Confidence:** 2

**Summary:**

This paper addresses the problem of the lack of reliable uncertainty estimates in semi-supervised node classification tasks using Graph Neural Networks.

This paper shows that nodes with the same label as the ego node play a critical role in the non-conformity scores of the ego node.

The authors propose a method to aggregate the non-conformity scores based on
feature similarity and structural neighborhood, to improve the efficiency of prediction sets and singleton hit ratio.

**Strengths:**

1. This paper is well-motivated, with clear motivation that nodes with high feature similarity or direct connections tend to have the same label.
2. This paper provides theoretical guarantee that the proposed method can consistently generate a smaller prediction set than basic non-conformity scores functions while maintaining the marginal coverage rate.
3. The authors provide adequate empirical analysis, including ablation studies and comparisons with state-of-the-art methods.

**Weaknesses:**

1. While the paper demonstrates the effectiveness of the proposed method on various datasets, it lacks a detailed analysis of the scalability / computation cost of the algorithm. It would be interesting to see how the method scales with the number of nodes and edges.
2. The focus on transductive learning in the paper limits its applicability to inductive learning scenarios, which are common in real-world classification tasks.
3. It would be interesting to see the discussion and experiments of the proposed method on heterphily graphs, where nodes with different labels are more likely to be connected.

**Questions:**

How will the proposed method perform on  heterphily graphs?

**Limitations:**

The authors discussed the limitations on transductive settings.

N/A potential negative societal impact.

---

> ### Author Rebuttal · Authors · 2024-08-07
>
> We appreciate the reviewer for the insightful and detailed comments. Please find our response below:
>
> > **1. Scalability / Computation cost [W1]**
>
> Thank you for your valuable feedback. The time complexity of SNAPS is primarily determined by the computation of corrected scores. In this work, we use one-hop nodes and nodes with high feature similarity to correct the ego node. In the transductive setting, this complexity applies to the entire graph. Consequently, one-hop generalization requires $\mathcal{O}(E)$ runtime, and K-NN generalization requires $\mathcal{O}(NM)$, where $E$ is the number of edges, $N$ is the number of test nodes, $M$ is the number of nodes sampled to correct the scores of test nodes, with $M\ll N$ for large graphs. Finally, the time complexity of SNAPS is $\mathcal{O}(E+NM)$.
>
> **Time complexity of k-NN.** Calculating pairwise similarities is inherently parallelizable, which makes k-NN significantly more efficient. Additionally, there have been some approximation methods that could be used to significantly speed up the computation for large graphs, such as NN-Descent [1] that can be easily implemented under MapReduce to empirically achieve an approximate k-NN graph in $\mathcal{O}(N^{1.14})$.
>
> > **2. Inductive learning [W2]**
>
> In the context of inductive learning, exchangeability is not maintained as changes in the graph affect the calibration nodes' conformity scores [2, 3]. Therefore, our primary focus is on transductive learning. Additionally, we present experimental results in the inductive scenario using CoraML dataset, illustrated in Figure 4 of the attachment. The results demonstrate that SNAPS generally achieves **valid coverage with comparable set sizes** in the inductive setting.
>
>
> > **3. Performance on heterophilous graphs [W3 & Q1]**
>
>
> We thank the reviewer for this intriguing question. Following the reviewer's advice, we conduct the experiments on two common heterophilous graph benchmarks. The benchmarks' details are as follows:
>
> | Datasets | Nodes | Features | Edges | Classes | Homophily Ratio |
> | :--: |:--: |:--: |:--: |:--: |:--: |
> | Chameleon | 2,277 | 2,325 | 36,101 | 5 | 0.23 |
> | Squirrel | 5,201 | 2,089 | 217,073 | 5 | 0.22 |
>
> FSGNN [4] is used as the GNN model, and we adopt the dataset splits from Geom-GCN [5], i.e. splitting nodes into 60%/20%/20% for training/validation/testing. To evaluate the performance of the CP methods, we divide the test set equally into the calibration and evaluation sets. For DAPS, we set $\lambda=0.5$. For SNAPS, we set $\lambda=0.5, \mu=0$ and $k=20$, i.e., neglecting the structural neighborhood. To construct a cosine similarity-based k-NN graph for these two heterophilous datasets, we utilize the embeddings from FSGNN. In the following tables, empirical results at $\alpha=\{0.10, 0.15\}$ are presented respectively:
>
> | | FSGNN |Coverage |  | |Size |  | |
> | :--: |:--: |:--: |:--: |:--: |:--: |:--: |:--: |
> |Datasets | Accuracy | APS | DAPS | SNAPS | APS | DAPS | SNAPS
> |Chameleon| 78.09±0.93 | 0.904 | 0.905 | 0.904 | 1.95 | 2.75 | **1.70** |
> |Squirrel| 73.72±2.19 | 0.900 | 0.897 | 0.900 | 2.41 | 3.05 | **2.27**|
>
> | |Coverage |  | | Size |  | |
> | :--: |:--: |:--: |:--: |:--: |:--: |:--: |
> |Datasets | APS | DAPS | SNAPS | APS | DAPS | SNAPS
> |Chameleon| 0.850 | 0.853 | 0.851 | 1.62 | 2.23 | **1.32** |
> |Squirrel| 0.851 | 0.848 | 0.850 | 1.89 | 2.37 | **1.64**|
>
> We can observe that SNAPS still shows **consistent superiority** over the baselines on heterophilous networks, which demonstrates its weak dependence on homophily.
>
> [1] Efficient K-Nearest Neighbor Graph Construction for Generic Similarity Measures. WWW'11.
>
> [2] DAPS: Conformal Prediction Sets for Graph Neural Networks. ICML'23.
>
> [3] Uncertainty Quantification over Graph with Conformalized Graph Neural Networks. NeurIPS'23.
>
> [4] Improving Graph Neural Networks with Simple Architecture Design. arxiv'21.
>
> [5] Geom-GCN: Geometric Graph Convolutional Networks. ICLR'20.

---

> > ### Comment · Reviewer_xCn2 · 2024-08-12
> >
> > Thank the authors for the extensive rebuttal, addressing most of my concerns.
> > I raise my score to 6.

---

> > > ### Author Response · Authors · 2024-08-12
> > > **Thank you for increasing the score**
> > >
> > > We are pleased that most of your concerns have been addressed. We sincerely thank you for raising your score.

---

### Official Review · Reviewer_3cJc · 2024-07-12

**Soundness:** 3
**Presentation:** 3
**Contribution:** 3
**Rating:** 6
**Confidence:** 3

**Summary:**

This paper introduces a novel algorithm, SNAPS, which enhances conformal predictions by aggregating non-conformity scores based on feature similarity and structural connections. Extensive experiments validate SNAPS' effectiveness, demonstrating its ability to produce more compact prediction sets with higher singleton hit ratios while maintaining rigorous finite-sample coverage guarantees.

**Strengths:**

1. The paper is clearly written and well-structured, facilitating easy comprehension.
2. It offers a new approach by utilizing similarity measurements based on node features to implement Conformal Prediction in node classification tasks, supported by a comprehensive theoretical analysis.
3. The paper conducts thorough experiments to test the validity of the proposed method, showing consistent improvements across various metrics.

**Weaknesses:**

1. Computing pairwise similarity is computationally demanding, especially for large-scale graph data.
Although the author notes in Appendix B. 1 that a subset was sampled to reduce computation costs, no details are provided on the sampling method or the size of the sample.

2. According to Figure 1(b), the difference in feature similarity between identical and different labels is minor, which does not convincingly justify the necessity of using feature similarity as an additional calibration method.

3. As shown in Figure 4, the success of this method hinges on the empirical selection of $\lambda$ and $\mu$, which restricts its broader application.

**Questions:**

1. Figure 1c shows that the number of nodes with the same label at the $k$-th nearest neighbors decreases as $K$ increases, but lines 272 -273 claim that more nodes with the same label are selected to enhance the ego node as $k$ increases. Does the explanation in lines 272-273 conflict with the observation in Figure 1c?
2. As shown in Table 9, the experiments were conducted on homogeneous datasets. Can this approach be extended to heterogeneous datasets? Will using high-similarity nodes for calibration remain effective in such settings?

**Limitations:**

See the weakness

---

> ### Author Rebuttal · Authors · 2024-08-07
>
> We appreciate the reviewer's insightful feedback. Please find our response below:
>
> > **1. Sampling method for large-scale dataset [W1]**
>
> To reduce the computation burden, we utilize a random subset from the original nodes (80,000 / 2449029, OGBN Products). Despite its simplicity and limited access to full datasets, the proposed SNAPS still establishes superior performance without high cost.
>
>
> > **2. Necessity of using feature similarity as an additional calibration method [W2]**
>
> Sorry for the misunderstanding caused by the lack of data precision. The features are high-dimensional and sparse in the vector space, leading to minor differences between similarity results. Therefore, we multiply the results shown in Figure 1(b) by 1000 and calculate the difference between the similarity of identical labels and that of different labels in each row, as shown in the following table:
>
> |class|0|1|2|3|4|5|6|
> |:-:|:-:|:-:|:-:|:-:|:-:|:-:|:-:|
> |0|0|25.90|31.38|36.27|35.76|19.08|20.48|
> |1|14.90|0|22.83|23.68|22.33|9.54|11.26|
> |2|22.32|24.76|0|33.78|29.37|20.06|17.36|
> |3|9.75|8.16|16.32|0|12.26|11.82|8.74|
> |4|10.08|7.64|12.75|13.10|0|9.09|4.68|
> |5|15.56|17.02|25.60|34.82|31.25|0|17.40|
> |6|26.01|27.79|31.96|40.80|35.90|26.45|0|
>
> The table demonstrates that the **relative difference in feature similarity between the identical and different labels is significant**.
>
> > **3. Choice of hyper-parameters $\lambda$ and $\mu$ [W3]**
>
> In the manuscript, SNAPS uses a hold-out dataset to tune the hyper-parameters, which is a common practice [1,2]. However, **there exist good default hyper-parameters for SNAPS** on most datasets, i.e., $\lambda=\mu=1/3, k=20$, whose experimental results at $\alpha=0.05$ are shown in the following:
>
> |||APS/RAPS/DAPS/SNAPS||
> |-|-|-|-|
> |Dataset|Coverage|Size|SH|
> |CoraML|0.950/0.958/0.957/0.951|2.50/2.62/2.32/**1.74**|43.09/27.34/44.52/**54.11**|
> |PubMed|0.950/0.968/0.967/0.950|1.82/2.10/2.09/**1.61**|33.39/14.66/23.27/**44.11**|
> |CiteSeer|0.951/0.950/0.952/0.950|2.41/2.69/2.16/**1.90**|48.53/35.37/55.40/**58.22**|
> |CS| 0.950/0.953/0.954/0.950|2.04/1.31/1.33/**1.13**|64.32/66.91/74.91/**85.21** |
> |Physics|0.951/0.962/0.962/0.950|1.39/1.44/1.28/**1.07**|72.44/62.22/77.65/**88.58**|
> |Computers|0.950/0.950/0.951/0.950|3.01/3.04/2.30/**2.01**|29.21/9.87/42.19/**45.98**|
> |Photo|0.949/0.950/0.950/0.950|1.90/1.81/1.56/**1.30**|54.86/47.27/67.57/**79.50** |
>
> The results show that SNAPS, with default hyper-parameters, outperforms RAPS and DAPS, even though both baseline methods are tuned on a hold-out dataset.
>
> > **4. Conflict between the explanation in lines 272-273 and the observation in Figure 1c [Q1]**
>
> Thank you for pointing the conflict out. There is indeed the ambiguity for the explanation in lines 272-273 of the manuscript. For clarity, we rephrase the statement as follows:
>
> Figure 4(a) and Figure 4(b) show that the performance of SNAPS significantly improves as k gradually increases from 0.  This improvement occurs because the increasing nodes with the same label are selected to enhance the ego node. Subsequently, as k continues to increase, the performance of SNAPS tends to stabilize.
>
>
> > **5. Extension to heterophilous datasets [Q2]**
>
> we guess your concern is about extension to heterophilous datasets, where nodes with different labels tend to be linked [3], because there may be inconsistencies in the types of nodes in heterogeneous datasets. To verify the efficiency of SNAPS on heterophilous graphs, we conduct the experiments on two common heterophilous graph benchmarks, as shown below:
>
> |Datasets|Nodes|Features|Edges|Classes|Homophily Ratio|
> |-|-|-|-|-|-|
> |Chameleon|2,277|2,325|36,101|5|0.23|
> |Squirrel|5,201|2,089|217,073|5|0.22|
>
> For the experiment setting, we choose FSGNN [4] as the GNN model and follow the dataset splits of Geom-GCN [5], i.e. splitting nodes into 60%/20%/20% for training/validation/testing. To evaluate the performance of the CP methods, we divide the test set equally into the calibration and evaluation sets. For DAPS, $\lambda=0.5$. For SNAPS, $\lambda=0.5, \mu=0$ and $k=20$, i.e., SNAPS neglecting structural neighborhood. To construct a cosine similarity-based k-NN graph for these two heterophilous datasets, we utilize the embeddings from FSGNN. In the following tables, empirical results at $\alpha=\{0.10, 0.15\}$ are presented respectively:
>
> ||FSGNN|Coverage|||Size|||
> |-|-|-|-|-|-|-|-|
> |Datasets|Accuracy|APS|DAPS|SNAPS|APS|DAPS|SNAPS|
> |Chameleon|78.09±0.93|0.904|0.905|0.904|1.95|2.75|**1.70**|
> |Squirrel|73.72±2.19|0.900|0.897|0.900|2.41|3.05|**2.27**|
>
> ||Coverage|||Size|||
> |-|-|-|-|-|-|-|
> |Datasets|APS|DAPS|SNAPS|APS|DAPS|SNAPS|
> |Chameleon|0.850|0.853|0.851|1.62|2.23|**1.32**|
> |Squirrel|0.851|0.848|0.850|1.89|2.37|**1.64**|
>
>
>
> [1] DAPS: Conformal Prediction Sets for Graph Neural Networks. ICML'23.
>
> [2] Uncertainty sets for image classifiers using conformal prediction. ICLR'21.
>
> [3] Graph Neural Networks for Graphs with Heterophily: A Survey. arxiv'22.
>
> [4] Improving Graph Neural Networks with Simple Architecture Design. arxiv'21.
>
> [5] Geom-GCN: Geometric Graph Convolutional Networks. ICLR'20.

---

> > ### Comment · Reviewer_3cJc · 2024-08-11
> >
> > Thank you for addressing most of my concerns. As a result, I have updated my score to 6.

---

> > > ### Author Response · Authors · 2024-08-11
> > > **Thank you for increasing the score**
> > >
> > > We appreciate the valuable suggestions and feedback from the reviewer. We are also glad that most of your concerns have been addressed. Thanks again for increasing the rating!

---

### Official Review · Reviewer_aWFN · 2024-07-12

**Soundness:** 2
**Presentation:** 3
**Contribution:** 2
**Rating:** 6
**Confidence:** 5

**Summary:**

The authors propose a new score function for conformal predictions on graphs. Given any baseline score the new score is aggregated based on the neighbors in the given graph; and the neighbors from a secondary kNN graph constructed based on the similarity between input features. The approach is motivated by noticing that augmenting the neighborhood of a node with other nodes from the same class improves performance.

**Strengths:**

The approach is well motivated and the proposed score is simple and intuitive which is a pro.

While I think the experimental evaluation can be improved (see weaknesses and questions) the experiments that are carried out are well described, thorugh, and help to support the claims made in the paper.

**Weaknesses:**

The calibration set size of $\min\\{1000, |V_{calib} \cup V_{test}|/2\\}$ seems problematic. For example, for Cora with 20 labels per class we have 20*7=140 labels for training/validation set. This means that a calibration set with size 1000 has an order of magnitude more labels. In practice it is much more likely to use most of the labels for training rather than calibration. At the very least results should be reported where the calibration set size is the same as the training/validation set size. Similarly, for ImageNet, equally dividing the data into a calibration set and a test set is not realistic.

In practice we either need to use fixed hyper-parameters or split the calibration set into 2 subsets: one for calibrating and one for tuning h-params. The authors do not discuss this issue (see also question 5).

The similarity graph is constructed based on a single heuristic (cosine similarity between node features). Considering other heuristics would be interesting, especially ones that also incorporate structure information and not only feature information.


Given the simplicity of the approach (which is a plus) the experimental evaluation should be strengthend (see questions).
I am willing to increase my score if the authors adequately address my questions.

**Questions:**

1. In Figure 1a) you show that the set size decreases as you increase the number of nodes with the same (oracle) label. This is effectively adding additional edges between nodes from the same class before the aggregation, increasing the homophily. This will likely increase the accuracy of the underlying model which non-surprisingly leads to reduced set size. How does the accuracy change if you e.g. take the argmin of the aggregated APS scores as a prediction or e.g. do vanilla GNN prediction on the augmented graph?
2. CF-GNN (Huang et al., 2023b) can in principle learn to do a similar aggregation to the one your propose. Can you please compare with them?
3. How does the performance of SNAPS (and the baselines) change as you vary the calibration set size? Importantly, also for small (realistic) sizes.
4. How does the performance of SNAPS (and the baselines) change as you vary the significance level $\alpha$?
5. What are the optimal h-params for different datasets and is there a good default value that works for most datasets? Relatedly, how does Figure 4c and 4d look like for other datasets?
6. Have you considered other similarity metrics?

**Limitations:**

The approach is likely to only work for graphs that have homophily (similar to DAPS). While this often holds for graphs of interest in practice, clearly highlighting this as a limitation would be appreciated.

---

> ### Author Rebuttal · Authors · 2024-08-07
>
> We deeply appreciate the valuable comments and will incorporate these suggestions into the final version. We are certain they will substantially improve the presentation of our work. Please find our response below:
>
> > **1. Calibration set size [W1 & Q3]**
>
> Here, we provide effect analysis of various calibration set sizes:
>
> 1. **Same calibration set size as training set.** We conduct experiments where the calibration set size is the same as the training set size (20 per class). Moreover, the calibration set is equally split into two sets: one for tuning h-params of CP methods and one for conformal calibration. SNAPS employs fixed h-params, i.e., $\lambda=\mu=1/3$ (A detailed analysis of fixed h-params for SNAPS can be found in Response 2 [W2 & Q5]). Here are the average results on 7 datasets at $\alpha=0.05$:
>
> ||APS/RAPS/DAPS/SNAPS||
> |-|-|-|
> ||Size|SH|
> |Average|2.15/2.14/1.86/**1.54**|49.41/37.66/55.07/**65.10**|
>
> The detailed results can be found in Table 1 of the attachment.
>
> 2. **Small calibration set size.** The results of the average size are shown in the following table
>
> |Num.|50|100|200|300|400|
> |-|-|-|-|-|-|
> |APS|2.92|2.50|2.43|2.42|2.39|
> |RAPS|3.12|2.76|2.35|2.33|2.26|
> |DAPS|2.84|2.49|2.06|2.07|1.96|
> |SNAPS|**2.02**|**1.76**|**1.72**|**1.71**|**1.70**|
>
> The results above both show that **SNAPS consistently outperforms other methods under different calibration set size setups**.
> In Figure 1 from the attachment, more results on different datasets can be found.
>
> For the vision dataset ImageNet, we provide the average results across different models for SNAPS and APS in Table 2 of the attachment.
> Despite the small size of calibration set, SNAPS still outperforms APS for classification problems.
>
>
> > **2. Choice of hyper-params [W2 & Q5]**
>
> **Optimal h-params.** In the manuscript, we use a hold-out dataset to tune the h-params, which is a common practice [1]. Here, we report the mean of the optimal h-params for SNAPS on different splits of each dataset.
>
> |Datasets|CoraML|PubMed|CiteSeer|
> |-|-|-|-|
> |$\lambda$|0.32±0.19|0.24±0.22|0.39±0.21|
> |$\mu$|0.44±0.13|0.51±0.24|0.27±0.19|
>
> **Good default hyper-params.** Moreover, there exist good default hyper-params for SNAPS on most datasets, i.e., $\lambda=\mu=1/3$, which indicates that three components of SNAPS are equally proportioned. The experiment results supporting this conclusion can be found in Response [W1 & Q3].
>
> **How does Figure 4c and 4d look like for other datasets?** We provide experimental results of other datasets in Figure 3 of the attachment.
>
> > **3. Other methods for similarity graph construction [W3 & Q6]**
>
> To incorporate structural information, we can use self-supervised learning to obtain embeddings of nodes [2]. Then, we use the cosine similarity between node embeddings to construct the similarity graph. To validate this method, we conduct experiments on the self-supervised model GraphACL [2] and follow its experimental setup. The experimental results are as follows:
>
> |SNAPS with|Original/GraphACL|Original/GraphACL|
> |-|-|-|
> |Datasets|$\alpha=0.05$|$\alpha=0.10$|
> |CoraML|**1.68**/1.71|1.31/**1.28**|
> |PubMed|1.62/1.62|1.35/**1.31**|
> |CiteSeer|1.84/**1.68**|1.39/**1.23**|
>
> The results demonstrate that the **similarity graph constructed based on self-supervised learning is applicable to SNAPS**.
>
> > **4. Effect of the aggregated APS scores / the augmented graph on the prediction accuracy [Q1]**
>
> Thank you for posing this insightful question. We take the argmin of the aggregated APS scores as a prediction and train vanilla GCN on the augmented graph. The prediction accuracy for these methods is as follows:
>
> |Methods/Datasets|CoraML|PubMed|CiteSeer|
> |-|-|-|-|
> |vanilla GCN|81.48|77.40|83.90|
> |argmin of DAPS|81.25|78.70|83.88|
> |argmin of SNAPS|81.83|79.43|84.08|
> |augmented graph|77.50|75.89|74.31|
>
> The results indicate that **SNAPS slightly enhances prediction accuracy**, which may be one reason for SNAPS's effectiveness.
>
>
> > **5. Comparison with CF-GNN [Q2]**
>
> To compare with CF-GNN, we randomly select 20 nodes per class for training/validation and set the calibration set size to 1,000 [3]. The APS score serves as the basic non-conformity score. The average results are calculated from 10 GCN with each trial of 100 conformal splits. Moreover, we introduce a metric, i.e., **Time**, to evaluate the running time for each trial. As shown in the Table below, SNAPS outperforms CF-GNN in both metrics.
>
> |Datasets|Size with $\alpha=0.05$|Size with $\alpha=0.1$|Time|
> |-|-|-|-|
> |||(CF-GNN/SNAPS)||
> |CoraML|2.60/**1.68**|1.68/**1.31**|142s/**0.73s**|
> |PubMed|2.13/**1.62**|1.86/**1.35**|148s/**0.98s**|
> |CiteSeer|3.07/**1.84**|1.96/**1.39**|124s/**0.71s**|
>
>
> > **6. Different significance levels $\alpha$ [Q4]**
>
> Here are the average set size across 10 GCNs with each trial of 100 conformal splits for CoraML at different significance levels:
>
> |Method\alpha|0.05|0.07|0.09|0.10|0.12|0.14|0.16|
> |-|-|-|-|-|-|-|-|
> |APS|2.42|2.10|1.90|1.81|1.70|1.60|1.52|
> |RAPS|2.16|1.81|1.58|1.43|1.28|1.18|**1.10**|
> |DAPS|1.92|1.66|1.47|1.44|1.32|1.25|1.20|
> |SNAPS|**1.68**|**1.48**|**1.36**|**1.30**|**1.23**|**1.18**|1.14|
>
> The results demonstrate that SNAPS outperforms other baselines at most significance levels. Additionally, we provide detailed results on PubMed and CiteSeer in Figure 2 of the attachment.
>
>
> > **7. Limitation to homophily**
>
> We clarify that the proposed method does not depend on homophily. Specifically, we conduct experiments on heterophilous datasets and SNAPS still outperforms baseline methods under low homophily ratio. Details can be found in response 2 to Reviewer 8yy7.
>
> [1] DAPS: Conformal Prediction Sets for Graph Neural Networks. ICML'23.
>
> [2] Simple and Asymmetric Graph Contrastive Learning without Augmentations. NeurIPS'23.
>
> [3] Uncertainty Quantification over Graph with Conformalized Graph Neural Networks. NeurIPS'23.

---

> > ### Comment · Reviewer_aWFN · 2024-08-11
> > **Updated score**
> >
> > Most of my concerns have been addressed. I have increased my score to 6.

---

> > > ### Author Response · Authors · 2024-08-11
> > > **Thank you for raising your score**
> > >
> > > Thank you for checking our rebuttal and raising your score. We will incorporate the new results and explanations into the final version appropriately. Sincerely thanks for your valuable time on this paper!

---

### Official Review · Reviewer_8yy7 · 2024-07-13

**Soundness:** 3
**Presentation:** 3
**Contribution:** 3
**Rating:** 7
**Confidence:** 3

**Summary:**

The authors apply conformal prediction to graph neural networks by aggregating the non-conformity scores based on both one-hop neighbors and feature similarity. The framework is verified through various experiments on graph ML benchmark datasets, where it's shown to generate smaller prediction sets and higher singleton hit ratio (i.e. only the correct answer in the set).

**Strengths:**

This paper is clearly presented and the results are fairly intuitive. It builds upon DAPS by adding a feature similarity term in eq (4). The experimental section is comprehensive in terms of number of datasets, ablation studies and parameter analysis.

**Weaknesses:**

One suggestion is to add a discussion on the assumptions in Proposition 2 and a proof sketch (if possible). For example, what does "$\Delta$ very small" really mean and how reasonable is it?

**Questions:**

I think it's possible that this method could more strongly outperform DAPS in heterophilous networks due to the addition of the term that doesn't depend on neighbors. Is there any chance the authors have tested their methods on a heterophilous network or observed any dependency of the performance on the degree of homophily?

**Limitations:**

yes

---

> ### Author Rebuttal · Authors · 2024-08-07
>
> We appreciate the reviewer's recognition and valuable suggestions. Please find our response below:
>
> > **1. Discussion regarding the assumption in Proposition 2 [W1]**
>
> Thank you for the great suggestion. Here is our discussion regarding the assumption in Proposition 2.
>
> Given a data pair $(\boldsymbol{x},y)$ and a model $f(\boldsymbol{x})$, we define a predicted probability estimator as $\pi(\boldsymbol{x})\_y$, where $\pi(\boldsymbol{x})\_y:=\sigma(f(\boldsymbol{x}))$ represents the predicted probability for label $y$ and $\sigma(\cdot)$ is an activation function, such as softmax. Let $\boldsymbol{S}$ denote APS scores of nodes, then we have
> $$\begin{aligned}\boldsymbol{S}\_{ui}=\sum\_{j=1}^{|\mathcal{Y}|}\pi(\boldsymbol{x}\_u)\_j\mathbb{I}[\pi(\boldsymbol{x}\_u)\_j>\pi(\boldsymbol{x}\_u)\_i]+\xi\cdot \pi(\boldsymbol{x}\_u)\_i,\end{aligned}$$ where $\boldsymbol{S}\_{ui}\in[0,1]$ is the score corresponding to node $u$ with label $i$, and $\xi\in[0,1]$ is a uniformly distributed random variable. Let $E\_k[\boldsymbol{S}\_{ui}]$ be the average of scores corresponding to label $i$ of nodes whose ground-truth label is $k$. Suppose $T$ is the number of nodes whose ground-truth label is label $k$.
>
> **a.** If $\pi(\boldsymbol{x}\_u)\_i$ is the largest predicted probability for node $u$, then $E\_k[\boldsymbol{S}\_{ui}]=E\_k[\xi\cdot \pi(\boldsymbol{x}\_u)\_i]=E\_k[\pi(\boldsymbol{x}\_u)\_k]+E\_k[\xi\cdot \pi(\boldsymbol{x}\_u)\_i]-E\_k[\pi(\boldsymbol{x}\_u)\_k]$. Suppose the number of nodes satisfying this case is A.
>
> **b.** Otherwise, $E\_k[\boldsymbol{S}\_{ui}]\geq E\_k[\pi(\boldsymbol{x}\_u)\_k]+E\_k[\xi\cdot \pi(\boldsymbol{x}\_u)\_i]$. Suppose the number of nodes satisfying this case is B, where $A+B=T$. Therefore, summing up $E\_k[\boldsymbol{S}\_{ui}]$ for both cases, we have
> $$A\cdot E\_k[\boldsymbol{S}\_{ui}]+B\cdot E\_k[\boldsymbol{S}\_{ui}]\geq (A + B)\cdot (E\_k[\pi(\boldsymbol{x}\_u)\_k]+E\_k[\xi\cdot \pi(\boldsymbol{x}\_u)\_i])-A\cdot E\_k[\pi(\boldsymbol{x}\_u)\_k], $$
>
> i.e.,
> $$E\_k[\boldsymbol{S}\_{ui}]\geq E\_k[\pi(\boldsymbol{x}\_u)\_k]+E\_k[\xi\cdot \pi(\boldsymbol{x}\_u)\_i]-\frac{A}{T}\cdot E\_k[\pi(\boldsymbol{x}\_u)\_k].$$
>
> Let $\epsilon=\frac{A}{T}\cdot E\_k[\pi(\boldsymbol{x}\_u)\_k]$, which reflects the average error of misclassifying label $k$ as label $i$. Let $\eta$ be $1-\alpha$ quantile of APS scores with a significance level $\alpha$. Then we have
> $$\begin{aligned}\eta=(1-\alpha)\frac{1}{|\mathcal{Y}|}\sum\_{j=1}^{|\mathcal{Y}|} E\_j[\pi(\boldsymbol{x}\_u)\_j].\end{aligned}$$ We can set $\eta - E\_k[\boldsymbol{S}\_{ui}]=\Delta$. If $\Delta\leq 0$, then $\eta \leq E\_k[\boldsymbol{S}\_{ui}]$, which means $\hat{\boldsymbol{S}}\_{vi}>\eta$ holds in Subsection A.2 of the manuscript. If $\Delta > 0$, then we have
> $$0<\Delta=\eta - E\_k[\boldsymbol{S}\_{ui}]
> \leq \eta - E\_k[\pi(\boldsymbol{x}\_u)\_k]-E\_k[\xi\cdot \pi(\boldsymbol{x}\_u)\_i]+\epsilon.$$
>
> The upper bound of $\Delta$ is given here. In most cases, $\eta\leq(1-\alpha)E\_k[\pi(\boldsymbol{x}\_u)\_k]$ or $\eta\approx(1-\alpha)E\_k[\pi(\boldsymbol{x}\_u)\_k]$, and then $0<\Delta<-\alpha E\_k[\pi(\boldsymbol{x}\_u)\_k]-E\_k[\xi\cdot \pi(\boldsymbol{x}\_u)\_i]+\epsilon$. Since $\epsilon$ reflects the average error of misclassifying label $k$ as label $i$, '$\Delta$ very small' is reasonable.
>
>
> > **2. Performance on heterophilous networks and dependency of the performance on the degree of homophily [Q1]**
>
> We thank the reviewer for this intriguing question. To analyze the performance of SNAPS on heterophilous networks, we conduct the experiments on two common heterophilous graph benchmarks. The benchmarks' details are as follows:
>
> | Datasets | Nodes | Features | Edges | Classes | Homophily Ratio |
> | :--: |:--: |:--: |:--: |:--: |:--: |
> | Chameleon | 2,277 | 2,325 | 36,101 | 5 | 0.23 |
> | Squirrel | 5,201 | 2,089 | 217,073 | 5 | 0.22 |
>
> We choose FSGNN [1] as the GNN model. We adopt the dataset splits of Geom-GCN [2], i.e. splitting nodes into 60%/20%/20% for training/validation/testing. To evaluate the performance of the CP methods, we divide the test set equally into the calibration and evaluation sets. For DAPS, we set $\lambda=0.5$. For SNAPS, we set $\lambda=0.5, \mu=0$ and $k=20$, i.e., neglecting the structural neighborhood. To construct a cosine similarity-based k-NN graph for these two heterophilous datasets, we utilize the embeddings from FSGNN. In the following tables, empirical results at $\alpha=\{0.10, 0.15\}$ are presented respectively:
>
> | | FSGNN |Coverage |  | |Size |  | |
> | :--: |:--: |:--: |:--: |:--: |:--: |:--: |:--: |
> |Datasets | Accuracy | APS | DAPS | SNAPS | APS | DAPS | SNAPS
> |Chameleon| 78.09±0.93 | 0.904 | 0.905 | 0.904 | 1.95 | 2.75 | **1.70** |
> |Squirrel| 73.72±2.19 | 0.900 | 0.897 | 0.900 | 2.41 | 3.05 | **2.27**|
>
> | |Coverage |  | | Size |  | |
> | :--: |:--: |:--: |:--: |:--: |:--: |:--: |
> |Datasets | APS | DAPS | SNAPS | APS | DAPS | SNAPS
> |Chameleon| 0.850 | 0.853 | 0.851 | 1.62 | 2.23 | **1.32** |
> |Squirrel| 0.851 | 0.848 | 0.850 | 1.89 | 2.37 | **1.64**|
>
> We can observe that SNAPS still shows **consistent superiority** over the baselines on heterophilous networks, which demonstrates its weak dependence on homophily.
>
> [1] Improving Graph Neural Networks with Simple Architecture Design. arxiv'21.
>
> [2] Geom-GCN: Geometric Graph Convolutional Networks. ICLR'20.

---

> > ### Comment · Reviewer_8yy7 · 2024-08-14
> >
> > Thank you for the reply. I have also read the other reviews and comments, and will keep my positive score.

---

> > > ### Author Response · Authors · 2024-08-14
> > > **Thank you for your positive score and recognition**
> > >
> > > Thank you for your recognition and for keeping the positive score. We are glad that our responses addressed these concerns, improving the quality of this work.

---

### Author Rebuttal · Authors · 2024-08-07

We thank all the reviewers for their time, insightful suggestions, and valuable comments. We are certain that they will make our work more complete. We are glad and encouraged that reviewers find the method is **well-motivated** (aWFN, xCn2) and **theoretical** (3cJc, xCn2), our method is **simple** (aWFN) and **effective** (8yy7, 3cJc, xCn2), and the experiments are **extensive** (8yy7, aWFN, 3cJc, xCn2). Besides, reviewers recognize that the writing is **easy to follow** (8yy7, 3cJc). We provide point-by-point responses to all reviewers' comments and concerns.

**Performance of SNAPS on heterophilous graphs**. We note that all reviewers are interested in the performance of SNAPS on heterophilous graphs. Thus, we add experiments on two heterophilous datasets and provide the experimental results in reponses below. For example, on the dataset Squirrel, SNAPS reduce the average set size from 1.89 (APS) to 1.64 when $\alpha=0.15$. The results demonstrate that SNAPS still outperforms other baseline methods on heterophilous graphs.

In summary, our method may reduce the set size for homophilous and heterophilous graphs. We will present the analysis in the final version.

---

### Decision · Program_Chairs · 2024-09-25

**Decision:**

Accept (poster)

**Comment:**

This paper proposes a simple extension of DAPS by Zargarbashi, Antonelli, and Bojchevski for conformal prediction on graph neural networks, which is named as SNAPS. The key idea is that, in addition to aggregate the non-conformity scores of the neighbors, SNAPS also aggregate the non-conformity score of nodes with the same labels into the non-conformity score of the ego node. During aggregation, to account for nodes without known label, the authors construct a feature similarity graph to select nodes with high features similarity for aggregation, based on the homophily assumption.

There are several advantages of the proposed work. It is very simple as the additional operations are aggregations over the original neighborhood and the high-similarity neighborhood, respectively. And the empirical results do show improvements on the prediction set size and the singleton hit ratio.

The reviewers raised several concerns regarding the evaluation of SNAPS, including comparison with CF-GNN, evaluation on heterophilous graphs, parameter analysis with varying significance level or calibration set size, etc. The additional results in the discussion phase consistently show a better performance than the existing methods. One thing I would suggest is about SNAPS on heterophilous graph: the way the authors evaluate it is to use embedding from FSGNN to construct feature similarity graph. Though it is a simple workaround for the heterophilous graphs, it doesn't really solve the problem on heterophilous where you hope that the embedding would be a good embedding on heterophilous embedding. So it would be great to add the additional results on heterophilous graphs and discussion about how you approach it, as well as the limitations in the future version.

My major concern about this paper is regarding the theoretical analysis. Reviewer 8yy7 raised concern about the assumption in proposition 2, specifically the meaning of "very small". It would be helpful if the authors could add the response into the future version, as it is unclear what "very small" or "very high" means rigorously, which should be avoided in a more rigorous statement of a theorem.

Overall, the paper has its own limitations but it has its own merits that the proposed method is simple and does offers improvements. All reviewers unanimously hold positive thoughts about this paper. I recommend accepting the paper.